# TEMPORAL GENERALIZATION: A REALITY CHECK

**Divyam Madaan**[1]    **Sumit Chopra**[1,2,3]    **Kyunghyun Cho**[1,2,4]
Courant Institute of Mathematical Sciences, New York University[1]
Center for Data Science, New York University[2]
Grossman School of Medicine, New York University[3]
CIFAR LMB[4]
{divyam.madaan, sumit,kyunghyun.cho}@nyu.edu

## ABSTRACT

Machine learning (ML) models often struggle to maintain performance under distribution shifts, leading to inaccurate predictions on unseen future data. In this work, we investigate whether and under what conditions models can achieve such a generalization when relying solely on past data. We explore two primary approaches: convex combinations of past model parameters (*parameter interpolation*) and explicit extrapolation beyond the convex hull of past parameters (*parameter extrapolation*). We benchmark several methods within these categories on a diverse set of temporal tasks, including language modeling, news summarization, news tag prediction, academic paper categorization, satellite image-based land use classification over time, and historical yearbook photo gender prediction. Our empirical findings show that none of the evaluated methods consistently outperforms the simple baseline of using the latest available model parameters in all scenarios. In the absence of access to future data or robust assumptions about the underlying data-generating process, these results underscore the inherent difficulties of generalizing and extrapolating to future data and warrant caution when evaluating claims of such generalization.

## 1 INTRODUCTION

> *"Prediction is very difficult, especially about the future." – Niels Bohr*

Temporal generalization of machine learning (ML) models is challenging, but critically important. After being trained on retrospective data, these models are deployed in real-world applications, particularly in high-stake domains such as finance, healthcare, and autonomous systems, where the distribution of the data the model sees could drift over time. Failures in these contexts can lead to severe financial losses or pose significant risks to human safety. It is therefore crucial to develop and evaluate models under deployment scenarios that accurately reflect temporal evolution. Yet, despite significant progress and widespread adoption, ML models suffer from performance degradation on data collected after the training period (Jaidka et al., 2018; Luu et al., 2021; Nylund et al., 2024). This issue of *temporal performance degradation* is widespread, affecting even large-scale state-of-the-art models such as GPT-4 (Achiam et al., 2023) and Gemini (Team et al., 2023). Figure 1 (left) shows this challenge, illustrating a 30% increase in perplexity when a model trained on past news summarization data is evaluated on future months, compared to a model adapted with more recent data. Such degradation in temporal generalization can lead to real-world consequences. The substantial financial and computational costs associated with training these foundation models (Cottier et al., 2024) make frequent retraining economically impractical. This necessitates alternative strategies for maintaining model utility over time.

With complete knowledge of the underlying data-generating process, it would, in principle, be possible to construct a model that generalizes effectively to future data. However, in most practical settings, this assumption is overly idealized and rarely holds. Motivated by this limitation, this paper addresses the following central question:

> *Can we build a model generalizable to the future without any access to future*
> *data and imperfect knowledge of the data-generating process?*

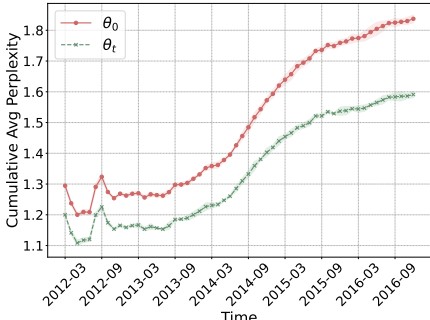 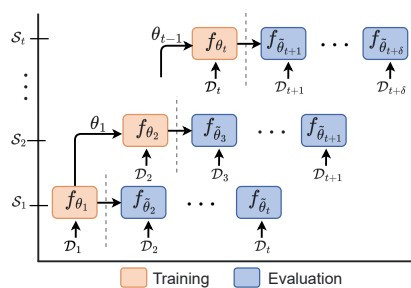

Figure 1: **(Left) Performance degrades over time.** The widening performance gap between a stale model trained once in January 2012 (red) and a monthly updated model (green) illustrates the decay in performance over time. The evaluation was conducted using data from March 2012 onward. **(Right) Temporal generalization framework.** Across sequential learning stages ($S_t$ on the y-axis), a model $f_{\theta_t}$ is trained on data $\mathcal{D}_t$ (orange) initialized with $\theta_{t-1}$ from the previous stage. This generates a sequence of historical parameters. This sequence is used to estimate future parameters $\widetilde{\theta}_{t+\delta}$, which are used to evaluate future data (blue) $\mathcal{D}_{t+\delta}$ for $\delta > 0$ (x-axis).

We adopt a stringent yet realistic setting to evaluate temporal generalization, as illustrated in Figure 1 (right). At a given time $t$, we have access to a sequence of model parameters (or checkpoints) $\{\theta_1, \ldots, \theta_t\}$ which resulted from training on historical data. Our objective is to leverage this sequence of checkpoints to estimate a new set of parameters, denoted $\widetilde{\theta}_{t+\delta}$, that will exhibit temporal generalization on unseen data $\mathcal{D}_{t+\delta}$ at a specific time into the future $t + \delta$ (where $\delta > 0$).

In addressing this question, we posit that any approach that only leverages retrospective data can be grouped into two categories: one that *interpolates* based on historical parameters and another that explicitly *extrapolates* towards the future.

**Parameter interpolation** (Ilharco et al., 2022; Yadav et al., 2024; Davari and Belilovsky, 2024; Jang et al., 2024; Dziadzio et al., 2024). This conservative yet intuitive approach restricts the search space for new parameters to the convex hull defined by past parameter checkpoints. We explore model merging (Ilharco et al., 2022; Yadav et al., 2024; Davari and Belilovsky, 2024) and a simple downscaling of the recent model parameters. The underlying intuition is twofold: parameters from the recent past might contain information relevant to the near future, and simple scaling might mitigate model overconfidence on unseen future data.

**Parameter extrapolation** (Nasery et al., 2021; Bai et al., 2023; Cai et al., 2024; Nylund et al., 2024). A more ambitious approach involves explicitly estimating future model parameters by extrapolating beyond the convex hull defined by the trajectory of past parameter values. This method operates under the hypothesis that the temporal evolution of model parameters, observable from historical training dynamics (Nylund et al., 2024), can inform predictions about future parameters, even in the absence of future data access. In particular, we investigate whether a Taylor series-based approximation can facilitate effective extrapolation of model parameters into the future.

We conducted a comprehensive empirical study comparing several methods representative of these two categories. Our evaluation spans a diverse array of temporal datasets and tasks, such as language modeling on evolving news corpora, news summarization and tag prediction, categorization of academic papers, land use classification from satellite imagery reflecting changes over several years, and gender prediction from historical yearbook photos spanning multiple decades.

There are significant challenges that complicate the rigorous evaluation of temporal generalization methods. Existing studies are limited to simplistic setups and lack real-world applicability due to high computational cost (Nasery et al., 2021; Bai et al., 2023; Cai et al., 2024), non-transparent hyperparameter selection (Bai et al., 2023; Cai et al., 2024; Nylund et al., 2024) and the use of future data during model adaptation (Nylund et al., 2024; Cha and Cho, 2024). Additionally, the inherent non-identifiability of deep neural networks (Hecht-Nielsen, 1990; Sussmann, 1992; Phuong and Lampert, 2020) yields non-linear parameter trajectories, which complicates both interpolation and extrapolation. Addressing these prevailing gaps, we provide the first holistic evaluation of these interpolation and extrapolation methods within large-scale settings.

The findings of this study highlight the intrinsic challenges of forecasting future model parameters based solely on historical trajectories, particularly in the absence of access to future data distributions or validated assumptions about the underlying data-generating process. Among the evaluated methods, the only one that consistently avoided performance degradation across datasets was the downscaling of recent model parameters. To better understand these outcomes, we conduct a detailed temporal analysis of parameter dynamics and discuss the implications of key design choices. We hope these insights will inform future research on the development and evaluation of temporally robust machine learning models. The code is available at `https://github.com/divyam3897/TG`.

**Contributions.** This work investigates temporal generalization through a large-scale evaluation of parameter interpolation and extrapolation. We adhere to the strict constraint of no access to the future and evaluate with multiple monthly and yearly temporal datasets and model architectures. Our principal empirical finding is that these parameter interpolation and extrapolation methods fail to improve, and often degrade, performance compared to using the most recent model. Our findings underscore the profound difficulty of predicting future model parameters solely from historical data under these stringent settings. This is because, without strong assumptions about how the data-generating process evolves over time or access to the future, the future can be arbitrarily different.

**Outline.** We begin by formalizing the problem of temporal generalization (§2). Next, we explore multiple approaches for parameter interpolation and extrapolation (§3), describing their underlying assumptions and associated difficulties (§4). Our experimental evaluation (§5) reveals that none of the methods investigated reliably achieve temporal generalization. Throughout our analysis, we identify key design principles and highlight directions for future research (§6).

## 2 THE PROBLEM OF TEMPORAL GENERALIZATION

Temporal generalization (or temporal domain generalization) (Nylund et al., 2024; Roth et al., 2024; Dai et al., 2024) refers to the ability of ML models to maintain performance over time. It can be viewed as a specific instance of classical domain generalization (DG) (Sun and Saenko, 2016; Arjovsky et al., 2019; Yue et al., 2019; Sagawa et al., 2020; Zhou et al., 2020; Zhang et al., 2021; Zhou et al., 2021; Yao et al., 2022b), where each timestamp corresponds to an ordered domain. The objective of temporal generalization is to generalize to future unseen timestamps. In this work, we focus on temporal generalization at a large scale, using models from 70M to 770M parameters on tasks like language modeling, news summarization, and classification without access to the future.

Consider a sequence of $T$ datasets, $\{\mathcal{D}_t\}_{t=1}^{T}$, where each $\mathcal{D}_t = \{(x_i^t, y_i^t)\}_{i=1}^{N_t}$ consists of $N_t$ data points collected at time $t$. At any current time $t$, we obtain model parameters $\theta_t$ by training on the current data segment $\mathcal{D}_t$. We assume access to the sequence of past model checkpoints $\{\theta_1, \ldots, \theta_t\}$. $\theta_t$ represents the latest checkpoint trained on the recent data $\mathcal{D}_t$. Our objective at time $t$ is to use historical checkpoints $\{\theta_1, \ldots, \theta_t\}$ to estimate new parameters $\widetilde{\theta}_{t+\delta}$ that will perform well on unseen future data $\mathcal{D}_{t+\delta}$ from a specific future time $t + \delta$ (where $\delta > 0$).

We focus on the streaming setting, which reflects real-world deployment scenarios. Often retraining on all historical data is computationally infeasible, and models must be continuously updated and evaluated. The estimation of $\widetilde{\theta}_{t+\delta}$ occurs without any access to future or past data (see Figure 1 (right)). This constraint of no future access provides a stringent and realistic test of a model's ability to *stand the test of time* using only the past.

**Infeasibility of non-linear approaches.** Ideally, one could capture temporal dynamics by training a sequence model, such as a RNN on a series of checkpoints (Bai et al., 2023) or by using compressed representations obtained by autoencoder (Cai et al., 2024). These approaches are impractical for large-scale models for two reasons. First, their computational complexity scales linearly with the number of parameters. It is intractable to predict a single large output layer for an entire language vocabulary. Second, they require fine-grained temporal data. Accurately modeling a non-linear parameter-time trajectory in a high-dimensional space of a large model requires a density of samples that can grow exponentially. This requirement is unmet by practical constraints, as publicly available datasets are typically too coarse (e.g., yearly snapshots with ten samples for ten-year data). Such parameter prediction techniques have been confined to small-scale models such as a 10-layer LSTM on toy datasets, inapplicable to widely used T5 (Raffel et al., 2020) and densenet-121 (Huang et al., 2017) architectures. We provide a more detailed discussion of related work in Appendix B.

Time Vectors (Nylund et al., 2024) is the only work that extrapolates language models by decomposing parameters into a time-varying language model component $\theta_t^{\text{LM}}$ and a fixed task-specific component $\theta_{\text{task}}$. While effective in some contexts, this approach has two major practical limitations. First, its success depends on a large, fixed pre-trained model to constrain the temporal updates, which is not applicable to settings without large-scale pre-training, such as the Wilds-Time datasets (Yao et al., 2022a). Second, it assumes access to unlabeled future data $\theta_{t+\delta}^{\text{LM}}$ and supervised future validation data to tune hyperparameters; a requirement often unmet in realistic scenarios. In the following section, we focus on methods that are scalable to large models, require no access to future data, and do not assume task-specific parameters to remain constant.

## 3   PARAMETER INTERPOLATION AND EXTRAPOLATION

In this section, we detail two approaches that leverage the past sequence of parameter checkpoints for temporal generalization: parameter interpolation and parameter extrapolation.

### 3.1   PARAMETER INTERPOLATION

Parameter interpolation constructs future parameters $\widetilde{\theta}_{t+\delta}$ as a weighted combination of parameters from historical model checkpoints.

**Model merging.** The general form of parameter interpolation involves creating an explicit weighted average of parameters from multiple past checkpoints. This approach, often referred to as model merging (Ilharco et al., 2022; Yadav et al., 2024; Davari and Belilovsky, 2024; Jang et al., 2024) consolidates diverse historical information to potentially improve generalization to future data $\mathcal{D}_{t+\delta}$ and mitigate over-reliance on the most recent checkpoint. Formally, the interpolated parameters are a convex combination of the past parameters:

$$\widetilde{\theta}_{t+\delta} = \sum_{i=0}^{t} \alpha_i \theta_i, \quad \text{where} \quad \alpha_i \geq 0 \text{ and } \sum_{i=0}^{t} \alpha_i = 1. \tag{1}$$

Here $\theta_i$ for $i \geq 1$ represents the model parameters trained on $\mathcal{D}_i$, and $\theta_0 = \mathbf{0}$ is defined as the zero vector for notational convenience. The hyperparameters $\alpha_i$ (for $i \geq 1$) are the weights determining the contribution of each checkpoint. These weights may be uniform (leading to simple averaging) or exponentially decaying (EMA) to assign greater importance to more recent checkpoints. Typically, $\alpha_0$ is set to zero, limiting the merging to the convex hull of $\{\theta_1, \ldots, \theta_t\}$.

While model merging (Ilharco et al., 2022; Yadav et al., 2024; Davari and Belilovsky, 2024; Jang et al., 2024) has shown promise when models are trained on similar tasks or datasets, merging models trained on distinctly different datasets (as can be the case with temporal data) is known to be challenging. This difficulty arises from high loss barriers between the parameter sets (Yamada et al., 2025). Consequently, existing merging or model editing techniques (Ilharco et al., 2023; Yadav et al., 2024; Sagawa et al., 2020; Wang et al., 2024; Fang et al., 2025) are not directly applicable to our strict problem setting, as they frequently rely on access to data from the target (i.e., future) distribution for model selection and validation. Dziadzio et al. (2024) recently showed that model averaging outperforms complex merging techniques (Yadav et al., 2024) over time. Their evaluation was restricted to curated datasets that did not exhibit strong temporal shifts and contained relatively similar data distributions over time. We revisit model averaging under naturally occurring temporal shifts and show that its effectiveness often diminishes as we evaluate further into the future.

**The recent model.** A straightforward baseline, and a specific instance of merging framework in Equation (1) is to deploy the most recent model $\theta_t$ trained on the current dataset $\mathcal{D}_t$. This corresponds to setting $\alpha_t = 1$ and $\alpha_i = 0$ for $i < t$ in the general framework. Since $\theta_t$ reflects the latest known data distribution, it is potentially relevant for near-future datasets. Nonetheless, previous studies (Luu et al., 2021; Lazaridou et al., 2021; Zhu et al., 2025; Dai et al., 2024) show that relying on recency can fail to generalize over time with some models collapsing to nearly random performance beyond their training cutoff. This stems from the fact that future data can change arbitrarily. In contrast to merely observing this degradation, we investigate proactive strategies to evaluate whether we can generalize better to the future. We observe that $\theta_t$, despite its simplicity, is often surprisingly competitive in time-evolving real-world settings.

**Parameter downscaling.** Another simplification of model merging is to focus only on the most recent model $\theta_t$, adjusting its overall magnitude by interpolating it towards the origin $\theta_0 = \mathbf{0}$. Given the parameters $\theta_t$, we derive parameters $\widetilde{\theta}_{t+\delta}$ using a single scaling hyperparameter $\alpha$:

$$\widetilde{\theta}_{t+\delta} \;=\; \alpha\,\theta_t \quad \text{with} \quad \alpha \in [0,1]. \tag{2}$$

This is a specific instance of the merging framework, where $\alpha$ is the weight for $\theta_t$, $1 - \alpha$ for $\theta_0$ and $\alpha_j = 0$ for $0 < j < t$. Downscaling reduces the parameter norm $\|\theta_t\|$ (if $\alpha < 1$) while preserving its direction (if $\alpha > 0$). The motivation is to prevent over-reliance on parameters optimized for the distribution at time $t$. This is inspired by empirical and theoretical observations, where parameter norms increase during training (Li and Arora, 2019; Ji and Telgarsky, 2020; Merrill et al., 2021; Nikishin et al., 2022; Dohare et al., 2024; Lewandowski et al., 2025), and larger norms correlate with sharper minima and reduced generalization (Foret et al., 2024; Zhao et al., 2022; Yashwanth et al., 2024). A larger norm reflects the model's strong reliance on the current data; for temporal generalization, reducing this norm might mitigate overfitting to the present time step. We show the existence of this phenomenon in Figure 4.

## 3.2 PARAMETER EXTRAPOLATION

A more direct, albeit ambitious, approach is to estimate the predictive distribution $f(x_{t+\delta}, \widetilde{\theta}_{t+\delta})$ for any future input $x_{t+\delta}$, where $\delta > 0$ represents the time increment into the future. To approximate $f(x_{t+\delta}, \widetilde{\theta}_{t+\delta})$, we consider how the parameters $\theta(t')$ might evolve as a differentiable function of time $t'$. Future parameters $\widetilde{\theta}_{t+\delta}$ can be related to current parameters $\theta_t$ via the approximation $\theta_{t+\delta} \approx \theta_t + \delta \cdot \theta'_t$, where $\theta'_t = d\theta(t')/dt'|_{t'=t}$ is the instantaneous rate of change of parameters at time $t$. The function $f$ could then be approximated via a Taylor expansion at $\theta_t$:

$$f(x_{t+\delta}, \widetilde{\theta}_{t+\delta}) \approx f(x_{t+\delta}, \theta_t) + \delta \cdot (\nabla_\theta f(x_{t+\delta}, \theta_t) \cdot \theta'_t) + \text{Higher-order terms.} \tag{3}$$

$\nabla_\theta f(x_{t+\delta}, \theta_t)$ is the gradient of $f$ with respect to its parameters $\theta$, evaluated at $\theta_t$ for input $x_{t+\delta}$. Nasery et al. (2021) is conceptually similar to this formulation, but it also requires specialized models that take $t$ as input. This is incompatible with standard pre-trained models. Computing Equation (3) is challenging because it requires computing $\nabla_\theta f(x_{t+\delta}, \theta_t)$ for every new future input $x_{t+\delta}$, which can be computationally prohibitive. To address this, we extrapolate the model parameters to an estimate $\widetilde{\theta}_{t+\delta}$. Once $\widetilde{\theta}_{t+\delta}$ is obtained, it can be used to make predictions $f(x_{t+\delta}, \widetilde{\theta}_{t+\delta})$. We formalize the parameter-time relationship locally using a Taylor approximation for $\theta(t')$ centered at the current time $t$. Since direct computation of $\theta'_t$ is infeasible without access to the underlying generative process of parameters, we approximate it using the most recent checkpoints $\theta_t$ and $\theta_{t-\Delta t}$:

$$\widetilde{\theta}_{t+\delta} \approx \theta_t + \alpha \cdot \theta'_t + \text{Higher-order terms}$$

$$\approx \theta_t + \alpha \frac{(\theta_t - \theta_{t-\Delta t})}{\Delta t}, \tag{4}$$

where $\Delta t > 0$ is the time interval between the past parameter checkpoints $\theta_t$ and $\theta_{t-\Delta t}$. The scalar $\alpha$ is distinct from the future time horizon $\delta$. It defines the extrapolation step size hyper-parameter, which determines how far along the estimated direction of change $\frac{(\theta_t - \theta_{t-\Delta t})}{\Delta t}$ we extrapolate. For example, if $\Delta t$ represents one discrete time unit of past observations, setting $\alpha = \Delta t$ would correspond to a linear extrapolation by an amount of change observed over one such past interval. A smaller positive $\alpha$ (e.g., $0 < \alpha < \Delta t$) yields a more conservative update along this trend.

Choosing $\alpha$ is critical for parameter extrapolation, which we discuss in the following section. While parameters appear to follow smooth trajectories in a low-dimensional space, their evolution in the high-dimensional space can be substantially more complex. We empirically show in Figure 5 that the optimal $\alpha$ often deviates significantly from our assumptions, sometimes even taking negative values, which suggests that interpolative approaches are more useful.

## 4 CHALLENGES WITH TEMPORAL GENERALIZATION

The relationship between model parameters $\theta$ and the function $f(\cdot, \theta)$ is complex due to the non-convex loss landscapes, which in turn results in issues related to identifiability. This poses challenges for the parameter interpolation and extrapolation methods presented in Section 3. This section outlines these challenges and describes our strategy to mitigate their impact on temporal generalization.

## 4.1 NON-CONVEXITY IN LOSS SURFACE RESULTING IN NON-IDENTIFIABILITY

When models are trained independently at different time steps $t$, the resulting parameters $\theta_t$ may reside in disparate basins of the loss surface. Parameter interpolation or extrapolation implicitly assumes that parameters from different time steps are connected and lie in the same basin or smoothly connected regions. Deep networks rarely satisfy this assumption. There often exist high barriers between different solutions, exacerbated by the lack of identifiability: different parameters $\theta^{(A)}$ and $\theta^{(B)}$ can produce the same functionally equivalent input-output mappings $f(\cdot, \theta^{(A)}) = f(\cdot, \theta^{(B)})$. Retraining yields different local minima due to factors such as weight permutations, initialization, and other random choices, leading to parameter sets that, while functionally similar, converge to different local minima (Hecht-Nielsen, 1990; Sussmann, 1992; Phuong and Lampert, 2020; Chen et al., 2024).

This presents a challenge for parameter analysis over time. For example, if the past parameters from Section 3.1 are distant from each other, interpolation and extrapolation between them might traverse regions of high loss, yielding poorly performing models. The finite difference $\theta_t - \theta_{t-\Delta t}$, central to Taylor expansion-based extrapolation (Equation (4)), may not capture a meaningful direction of change if the parameters are not aligned. Consequently, a naively constructed parameter-time trajectory $\theta(t)$ can contain unknown noise or discontinuities. This makes it difficult to discern any true underlying temporal structure suitable for reliable interpolation or extrapolation. We provide a conceptual illustration of this issue with synthetic data in Appendix A.

To address this challenge, we adopt sequential fine-tuning, a specific instance of continual learning (CL) for training models over time. Specifically, when training the model for time step $t$ on dataset $\mathcal{D}_t$, we initialize its parameters with $\theta_{t-1}$, the optimized parameters obtained from training on $\mathcal{D}_{t-1}$. The model is then fine-tuned:

$$\theta_t = \arg\min_{\theta_t} \sum_{(x_i^t, y_i^t) \in \mathcal{D}_t} \text{CE}\big(f(x_i^t; \theta_t), y_i^t\big) \quad \text{with } \theta_t \text{ initialized from } \theta_{t-1}. \tag{5}$$

By initializing from the previous solution, we show that consecutive parameters stay close to each other (Figure 7), aiding parameter interpolation and extrapolation. We remark that our goal of interpolation and extrapolation is different from prior interpolation studies (Hecht-Nielsen, 1990; Sussmann, 1992; Phuong and Lampert, 2020; Chen et al., 2024) that interpolate a path between two known model optima trained on the same dataset. We find a path to an unknown future model state, which is exceptionally difficult. This is because constructing a non-linear path would require strong, and typically unavailable, assumptions about the evolution of data distribution with time.

## 4.2 CHALLENGES IN HYPERPARAMETER TUNING

The efficacy of parameter interpolation and extrapolation methods heavily relies on selecting appropriate hyperparameters, such as the coefficients $\alpha$ in Equation (1), Equation (2), and Equation (4). In temporal generalization, this process presents a distinct challenge as hyperparameters must be chosen using only historical data. Any use of future data, which is reserved for evaluation is methodologically unsound. Such a practice would lead to overly optimistic performance assessments that do not reflect true generalization capabilities (Nylund et al., 2024; Cha and Cho, 2024).

We sequentially tune the hyperparameter $\alpha$ by emulating a deployment scenario. At each time step $t$, with current model parameters $\theta_t$ and data $\mathcal{D}_t$, we determine the $\alpha$ to generate $\widetilde{\theta}_{t+\delta}$. Particularly, we simulate the hyperparameter choice we would have made at the previous step and find the optimal $\alpha^*$ by assessing performance using a metric $\mathcal{L}$ on a validation subset of the current data, $\mathcal{D}_t^{\text{val}}$. This involves generating candidate parameters $\widetilde{\theta}_t(\alpha)$ using past parameters and a candidate value $\alpha^*$:

$$\alpha^* = \arg\min_{\alpha \in \mathcal{S}} \mathcal{L}\left(f\left(\cdot; \widetilde{\theta}_t(\alpha)\right), \mathcal{D}_t^{\text{val}}\right). \tag{6}$$

Here, $\widetilde{\theta}_t(\alpha)$ are the parameters generated for this validation. The set $\mathcal{S}$ represents the defined search space for $\alpha$, specific to the temporal generalization method (e.g., the interval $[0, 1]$ for downscaling, and $\mathbb{R}$ for extrapolation). This process is repeated as new data becomes available and the resulting $\alpha^*$ is then used with the current parameters $\theta_t$ to generate $\widetilde{\theta}_{t+\delta}$ for upcoming periods.

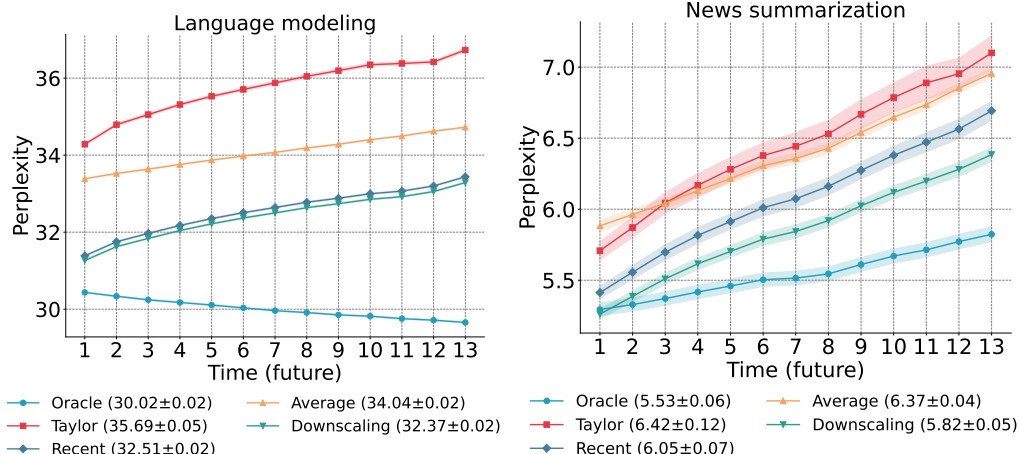

Figure 2: **Average perplexity comparison for T5-small model.** Results for language modeling are in the left figure, and those for news summarization are in the right. Every month, each method is evaluated over 12 future months (x-axis), with lower perplexity (y-axis) indicating better performance. Downscaling is the only method that did not lead to a decrease in performance. Oracle contains complete knowledge of the future, trained and evaluated on data from time $t$.

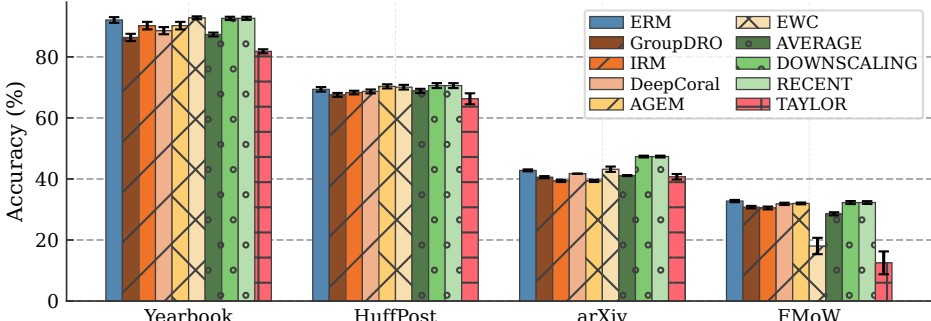

Figure 3: **Comparison of accuracy for Wilds-Time datasets.** We compare ERM (trained on union of data up to time $t$) with DG (GroupDRO (Sagawa et al., 2020), IRM (Arjovsky et al., 2019), Deep-Coral (Sun and Saenko, 2016)), CL methods (EWC (Kirkpatrick et al., 2017), AGEM (Chaudhry et al., 2019)), parameter interpolation (Average, Downscaling, Recent), and Taylor extrapolation over $\delta$ future months. No method consistently outperforms others across datasets. The details for the methods are provided in Appendix E.

## 5 EXPERIMENTAL RESULTS

We experiment with NewsRoom dataset (Grusky et al., 2018) to solve the language modeling and news summarization tasks using the T5-models (Raffel et al., 2020) in Figure 2. Additionally, we use Wilds-Time (Yao et al., 2022a) benchmark containing Yearbook (Ginosar et al., 2015), FMoW (Christie et al., 2018; Koh et al., 2021), HuffPost (Misra and Grover, 2021) and arXiv (Clement et al., 2019) datasets in Figure 3. As detailed in Appendix C, a primary constraint for temporal generalization is the lack of public datasets with fine-grained temporal resolution. Most benchmarks offer only a few coarse time points (e.g., yearly data), which is insufficient for meaningful extrapolation. The NewsRoom dataset is a benchmark that contains monthly granularity.

We measure temporal generalization with $\delta$-forward transfer (Lopez-Paz and Ranzato, 2017; Yao et al., 2022a), where we evaluate on $\delta$ future time-stamps for all the datasets. Details regarding datasets, evaluation metrics, hyperparameter settings and models are in Appendix C, Appendix D, and Appendix E, respectively. Additional results are in Appendix F. Takeaway from our experiments is that the inconsistent performance of benchmarked approaches across multiple datasets. Based on our evaluation, we present four primary findings:

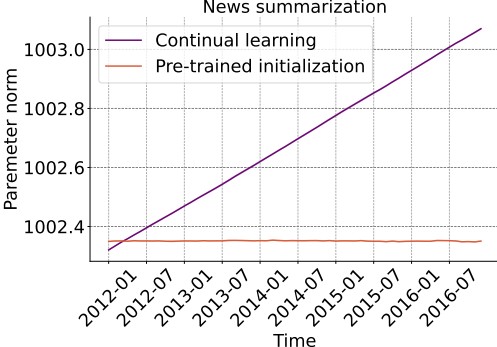

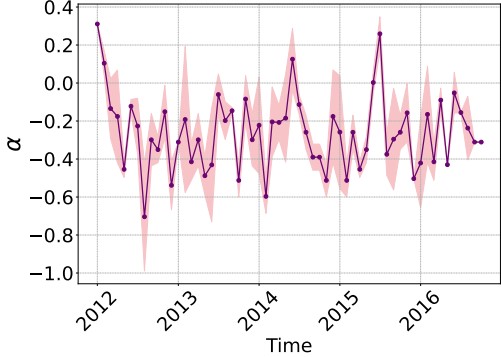

Figure 4: **Effect of downscaling.** The L2 norm of model parameters increases over time under continual learning, while it remains flat for models reinitialized at each step. Downscaling reduces this overconfidence and improves temporal generalization.

Figure 5: **Effect of extrapolation factor.** The optimal extrapolation factor $\alpha$ fluctuates with time and is often less than one or even negative. This indicates that sometimes interpolation is preferable over extrapolation for temporal generalization.

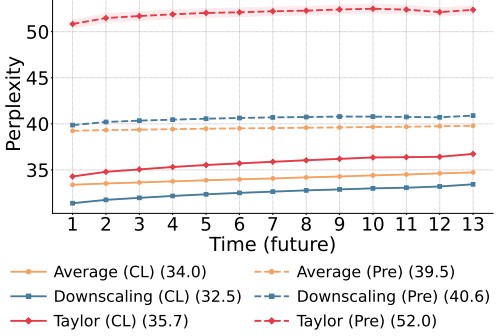

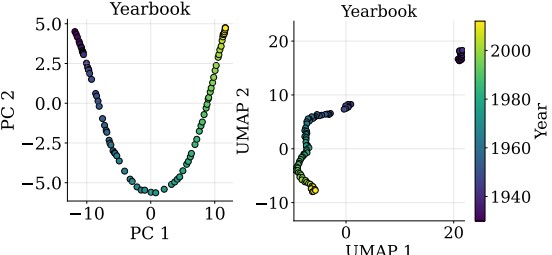

Figure 6: **Effect of CL.** For language modelling, CL (solid lines) improves perplexity compared to pre-trained model (dashed lines).

Figure 7: **Dimensionality reduction.** PCA (left) and UMAP (right) on models trained with yearbook dataset showing the change of parameters with time.

**1) No method consistently improves performance over the recent model.** Across all datasets, our results show that most methods degraded performance compared to using only the most recent model. The performance decline with model averaging s likely due to older parameters introducing noise as a result of distributional shifts over time. Another key observation is that downscaling the most recent parameters with a single scalar reduces perplexity for language modeling and news summarization, but gives sub-par performance on Wilds-Time datasets. As shown in Figure 4, the norm of the parameters consistently grows over time, potentially hurting generalization as discussed in Section 3. This observation supports the strategy of shrinking the parameter norm at inference ($\alpha < 1$), obtaining a similar (sometimes better) model performance for temporal generalization as it reduces the present model's overconfidence against an unpredictable future.

**2) Taylor-Series extrapolation underperforms compared to other methods.** Its performance was consistently lower than that of a recent model. This outcome contrasts with the successful extrapolation reported for Time-vectors by Nylund et al. (2024) due to the lack of access to the future data in our setting. To show that this simple approach does not work, we tuned the parameter $\alpha$ in Equation (4) using future validation data at each timestamp for the news summarization task (see Figure 5 for $\alpha$ trends). The optimal values for $\alpha$ were frequently less than one or negative. An optimal $\alpha < 1$ suggests that a dampened extrapolation is preferable, while a negative $\alpha$ indicates that adjusting in the direction opposite to the first-order estimated change yields better performance. This implies that interpolation is sometimes preferable over forward extrapolation for temporal generalization.

**3) Continual learning is important.** Figure 6 compares CL with training each timestamp model independently, initialized from a pre-trained T5-small checkpoint. The results demonstrate that CL improves forward transfer for both interpolation and extrapolation methods significantly. This rein-

forces our hypothesis that initializing each time step parameters with the previous one keeps parameters close to each other and preserves a consistent trajectory in parameter space. This smoothing effect is further illustrated by comparing the principal component projections of parameter trajectories: CL yields smooth, coherent paths (Figure F.14), contrasting sharply with the disjoint and noisy trajectories observed when models are trained independently from a pre-trained model (Figure F.15).

We highlight that Figure 6 shows that sequential fine-tuning is necessary for any extrapolation method to work. All methods perform better with a CL backbone than with a pre-trained initialization because the parameters for adjacent time steps must lie close to each other. Figure 4 shows that this is not a sufficient condition. While CL helps, it can also lead to an increase in the parameter norm, which is reduced by using downscaling as a simple method.

## 6 LIMITATIONS AND FUTURE DIRECTIONS

**Fundamental constraints on temporal generalization.** The challenge of temporal generalization is subject to fundamental theoretical constraints, notably those underscored by the No Free Lunch theorem. Without strong and empirically validated assumptions regarding the nature of temporal distribution shifts, no algorithm can universally guarantee optimal future performance against arbitrary changes. This principle highlights an inherent vulnerability of any method: an approach that performs well on certain future distributions may fail on others due to the inherently arbitrary and unpredictable nature of future distributional shifts.

**Future directions.** Given these theoretical constraints, a key future direction is to model the non-linear evolution of parameters with time by making explicit assumptions about how the data-generating process evolves over time. Our dimensional reduction analysis provides an empirical starting point in Figure 7, Figure F.13. We find a one-dimensional structure corresponding to time, but the interaction of this dimension with the full parameter space remains an open problem. As a promising first step, we begin by capturing this relationship with two approaches below:

**Learning the change (a)** considers learning a single offset $\theta^\Delta$ to capture parameter changes:

Table 1: Perplexity comparison of Taylor expansion with (a) and (b).

$$\min_{\theta^\Delta} \sum_t \sum_{\delta=0}^{\tau} \left\| \theta_t - \left(\theta_{t-\Delta t} + \theta^\Delta\right)\right\|_2 \; + \; \lambda \left\|\theta^\Delta\right\|_2, \qquad (7)$$

| (a) | (b) | Future (12 months) |
|-----|-----|--------------------|
|     |     | $6.05_{0.07}/13.06_{0.17}$ |
| ✓   |     | $6.09_{0.09}/13.07_{0.16}$ |
| ✓   | ✓   | $6.07_{0.09}/13.07_{0.17}$ |

where $\theta^\Delta$ represents the learned change in parameters. We include a regularization term $\left\|\theta^\Delta\right\|_2$ to find the minimum parameter change.

**Learning the coefficient (b)** allows the extrapolation to multiple time periods into the future by learning the coefficient as:

$$\min_{\theta^\Delta, \alpha, \beta} \sum_t \sum_{\delta=0}^{\tau} \left\| \theta_t - \left(\theta_{t-\Delta t} + \text{softplus}(\alpha\delta + \beta)\,\theta^\Delta\right)\right\|_2 + \lambda \left\|\theta^\Delta\right\|_2, \qquad (8)$$

where $\text{softplus}(\alpha\,\delta + \beta)$ provides a smooth, positive scaling factor that scales the magnitude of $\theta^\Delta$. Despite enforcing minimal parameter change and allowing for a time-varying scale, Table 1 shows that we underperform the most recent model. The challenge lies in understanding the interplay between millions of parameters. While parameter extrapolation seems appealing, real-world data needs a conservative approach for temporal generalization.

## 7 CONCLUSION

Our work highlights the challenges and pitfalls in temporal generalization, an area where prior work has made strong claims of generalization. We systematically investigated this problem through the perspectives of interpolation and extrapolation of parameters without access to future data. Our analysis across multiple datasets demonstrates the absence of a superior method compared to the most recent model, underscoring the nuanced and context-dependent nature of this problem. Our findings highlight that the key to temporal generalization is thus not to design new algorithms, but to identify the reasonable assumptions about how the data generating process evolves over time. Only by making those assumptions explicit can we hope to develop methods that generalize over time.

ACKNOWLEDGEMENT

This work was supported by the Institute of Information & Communications Technology Planning & Evaluation (IITP) with a grant funded by the Ministry of Science and ICT (MSIT) of the Republic of Korea in connection with the Global AI Frontier Lab International Collaborative Research, Samsung Advanced Institute of Technology (under the project Next Generation Deep Learning: From Pattern Recognition to AI), National Science Foundation (NSF) award No. 1922658, Center for Advanced Imaging Innovation and Research (CAI2R), National Center for Biomedical Imaging and Bioengineering operated by NYU Langone Health, and National Institute of Biomedical Imaging and Bioengineering through award number P41EB017183. The computational requirements for this work were supported by NYU IT High Performance Computing resources, services, and staff expertise and NYU Langone High Performance Computing Core's resources and personnel. This work was partly supported in part by the NYUAD Center for Interdisciplinary Data Science & AI (CIDSAI), funded by Tamkeen under the NYUAD Research Institute Award CG016. This content is solely the responsibility of the authors and does not represent the views of the funding agencies.

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

APPENDIX

**Organization.** In the appendix, we discuss the challenge of non-convexity for temporal generalization (Appendix A), the datasets used and their specifics (Appendix C), the evaluation metrics applied (Appendix D), the hyperparameter configurations for our experiments (Appendix E), and additional experimental results (Appendix F).

## A    CHALLENGE OF NON-CONVEXITY FOR TEMPORAL GENERALIZATION

We define a model's parameters $\theta_t$ as identifiable if distinct parameter values always yield distinct predictive functions:

$$f(\cdot; \theta_t^{(A)}) = f(\cdot; \theta_t^{(B)}) \implies \theta_t^{(A)} = \theta_t^{(B)}, \tag{9}$$

for any two parameterizations $\theta_t^{(A)}$ and $\theta_t^{(B)}$, where $f(\cdot; \theta_t)$ is the model's predictive function. If different learned parameters $\theta_t$ result in functionally identical models, the parameter-time function can be noisy, posing a significant challenge in extrapolating these parameters to $\widetilde{\theta}_{t+\delta}$. In this section, we highlight these issues within a controlled synthetic time-varying regression setup, contrasting linear and non-linear models.

**Synthetic Experimental Setup.** To isolate identifiability effects, we consider input features $\mathbf{x}_t \in \mathbb{R}^d$ sampled independently at each time step $t$ from $\mathbf{x}_t \sim \mathcal{N}(\mathbf{0}, \mathbf{I})$. The underlying true data-generating parameters $\theta_t^* \in \mathbb{R}^d$ evolve via a cubic polynomial:

$$\theta_t^* = \mathbf{a} + \mathbf{b}t + \mathbf{c}t^2 + \mathbf{d}t^3, \tag{10}$$

where $\mathbf{a}, \mathbf{b}, \mathbf{c}, \mathbf{d} \in \mathbb{R}^d$ are fixed coefficients. The observed continuous target $y_t$ is generated by:

$$y_t = \mathbf{x}_t^T \theta_t^* + \epsilon_t, \tag{11}$$

with $\epsilon_t \sim \mathcal{N}(0, \sigma_\epsilon^2)$ being i.i.d. Gaussian noise. We compare two modeling approaches:

1. **Linear Regression Model** predicts $y_t$ using a linear function of the inputs,

$$\widehat{y}_t = \mathbf{x}_t^T \theta_t. \tag{12}$$

2. **Non-linear Regression Model** uses a non-linear model, such as a Multi-Layer Perceptron (MLP) with parameters $\theta_t$:

$$\widehat{y}_t = f(\mathbf{x}_t; \theta_t). \tag{13}$$

For both models, we learn parameters $\theta_t$ from sequences $(\mathbf{x}_t, y_t)$ and evaluate on the future data.

**Identifiability and Extrapolation in Linear Models.** Linear regression yields parameters $\theta_t$ that consistently converge to the true parameters $\theta_t^*$. This implies that both the magnitude and the direction of $\theta_t^*$ are identifiable via $\theta_t$. Figure A.8 shows that the Taylor series expansion of the learned trajectory $\theta_t$ produces reliable estimates of future parameters $\widetilde{\theta}_{t+\delta}$. Consequently, the model achieves better temporal generalization, maintaining predictive accuracy on future data, as shown in Figure A.10.

**Identifiability Challenges and Extrapolation in Non-linear Models.** Non-linear models incur identifiability challenges with their parameters $\theta_t$. Even if we accurately learn the input-output relationship, the MLP parameters $\theta_t$ may not be unique. This lack of uniqueness stems from several factors: (i) *Symmetries*: Many non-linear architectures possess inherent symmetries (for example, permutation of hidden units in an MLP) where different $\theta_t$ yield identical functions. (ii) *Non-convex Optimization*: The optimization landscapes for these models are typically nonconvex, meaning the optimizers may converge to different local minima, each corresponding to distinct $\theta_t$ that are functionally similar on the training distribution but structurally different.

These identifiability issues imply that the specific learned parameter trajectory $\theta_t$ might be one of many possible trajectories, exhibiting noisy and unstable behavior over time. Figure A.9 shows the

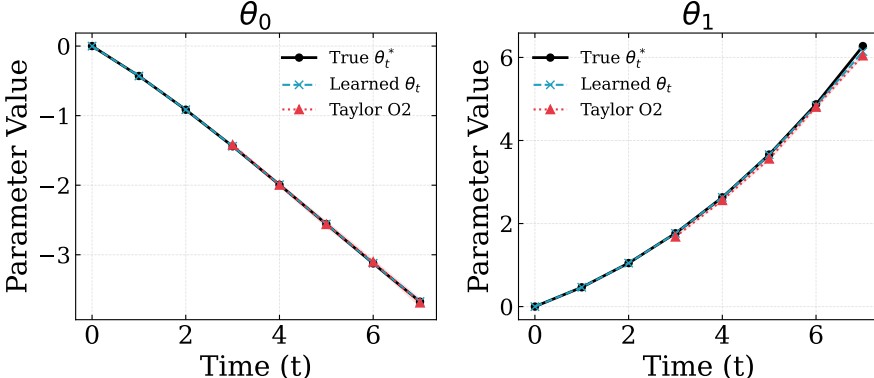

Figure A.8: Comparison of true underlying parameters $\theta_t^*$, learned parameters $\theta_t$ of a linear model and Taylor second-order extrapolated parameters $\tilde{\theta}_t$. The identical plots for the true parameters and extrapolated parameters illustrate effective parameter estimation.

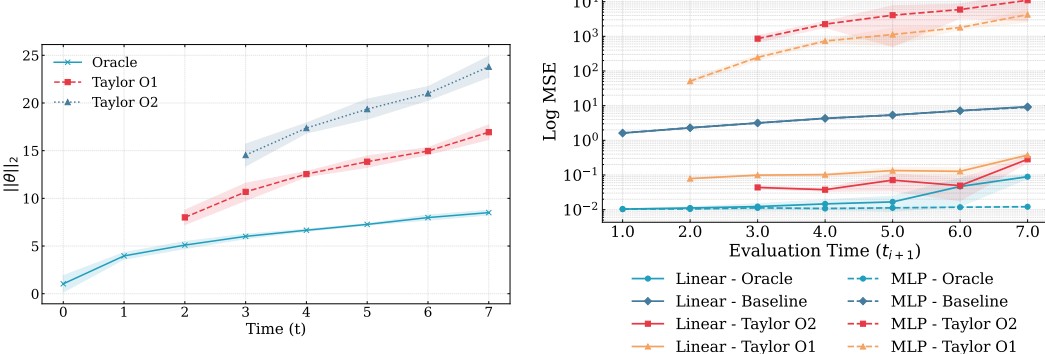

Figure A.9: **MLP Parameter Norm Evolution:** $\ell_2$ norm comparison of MLP Oracle, and extrapolated parameters using Taylor O1 (first-order) and O2 (second-order) approximations. The divergence in extrapolated norms highlights challenges in extrapolating MLP parameter trajectories across time.

Figure A.10: **Log MSE Extrapolation Performance:** One-step-ahead performance comparison for linear (solid lines) and MLP (dashed lines) models. The linear model with identifiable parameters allows for more effective Taylor extrapolation than MLP, whose parameter non-identifiability impedes performance. The oracle performanace of linear and MLP overlaps.

$\ell_2$ divergence of Taylor approximation from the true model. Consequently, directly extrapolating such a $\theta_t$ trajectory to obtain future parameters $\tilde{\theta}_{t+\delta}$ becomes highly unreliable compared to a linear model as shown in Figure A.10. This inability fundamentally limits the ability of temporal generalization and contrasts sharply with well-specified linear models where parameter estimate $\tilde{\theta}_{t+\delta}$ is meaningful.

# B  RELATED WORK

**Temporal Generalization.** Primary challenges for temporal generalization methods are computational scalability and data scarcity, which render them impractical for large-scale models. Approaches that train auxiliary models over the parameters of a network, such as recurrent networks (Bai et al., 2023) or autoencoders (Cai et al., 2024), exemplify this problem. The recurrent method of Bai et al. (2023) has a complexity of $\mathcal{O}(Nd+C)$, with $N$ being the parameter count of the predictive model, $d$ the width of the last hidden layer, and $C$ the parameter count of preceding layers. Its linear scaling with $N$ is intractable for large models, especially those with high-dimensional outputs such as a language vocabulary. Similarly, training an autoencoder on the full parameter set (Cai et al.,

2024) of widely used architectures is computationally infeasible. The coarse granularity of available temporal data, often limited to yearly checkpoints, makes learning the underlying dynamics challenging and undermines the justification for such data-intensive approaches. The approach in Nasery et al. (2021) models the function's output with respect to a time input, which necessitates intrusive architectural changes and full model fine-tuning. For practical applications, we avoid the need for expensive auxiliary models, architectural modifications, and reliance on fine-grained temporal data.

**Online learning.** Our problem setting also relates to, yet differs significantly from, online learning. Online learning emphasizes rapid adaptation to streaming data, where a model sequentially predicts, observes an outcome, and updates to minimize cumulative regret over time, focusing on how quickly it can converge to the best dynamic solution (Zinkevich, 2003; Duchi et al., 2011; Rakhlin and Sridharan, 2013; Hazan et al., 2016). In contrast, temporal generalization assesses a model's performance on unseen future data using parameters at time $t$, without any adaptation during this future deployment. Thus, while online learning is concerned with the efficiency of continuous adaptation within an immediate predict-observe-update loop, our focus is on the *a priori* generalization to future distributions. The challenge in temporal generalization lies in necessitating assumptions about how the future might relate to the past, a consideration less central to the regret minimization framework of online learning.

## C  DATASETS

This section details the datasets used in our work. These datasets were selected primarily for their distinct temporal characteristics. Collectively, they span a diverse range of tasks and data modalities, which facilitates a thorough investigation of model behavior over time:

- **NewsRoom** (Grusky et al., 2018): This dataset is a large corpus comprising approximately 9 million news articles published between 2009 and 2016. Following Luu et al. (2021); Nylund et al. (2024), we partition the data by month and year to evaluate both language modeling and news summarization tasks. For language modeling, we specifically focus on the 2012–2016 period, utilizing the English subset of the WMT news dataset (Barrault et al., 2020). From this WMT subset, we sample approximately 7.1 million tokens from articles each month for training and 700k–720k tokens for testing per month. WMT training and testing splits for August 2012 and May 2016 were unavailable. For the news summarization task (NewsSum), our setup is based on the original task defined by Grusky et al. (2018); Nylund et al. (2024) with the post-processing steps detailed by Luu et al. (2021). To the best of our knowledge, NewsRoom is distinct in offering data at this scale for monthly temporal evaluations, providing approximately 60 months (5 years) of continuous data from 2012–2016 suitable for this kind of analysis for temporal generalization.

- **WILDS-Time** (Yao et al., 2022a): This benchmark comprises multiple, temporally-structured datasets spanning diverse modalities. In this work, we focus on four specific datasets from this benchmark: the image-based datasets Yearbook and FMoW, and the text-based datasets ArXiv and HuffPost.

  - *Yearbook* (Ginosar et al., 2015): This dataset consists of 37k grayscale portrait photographs from U.S. high school yearbooks between 1930 and 2013. The task is to predict binary gender labels from facial features. Faces and fashion styles evolve significantly over time, making this a useful benchmark for evaluating visual domain shifts.

  - *HuffPost* (Misra and Grover, 2021): This dataset consists of news headlines published between 2012 and 2018, each labeled with one of several topic categories (e.g., politics, entertainment, technology). The task is to classify headlines, thereby simulating topic identification challenges under the influence of evolving media discourse and shifting language use over time.

  - *arXiv* (Clement et al., 2019): Comprising paper titles submitted to arXiv from 2007 to 2022 (2M examples), the goal is to predict the subject category (e.g., CS.LG, math.AP). The temporal challenge stems from the gradual evolution of scientific language, occasionally marked by more abrupt shifts due to the emergence of new research fields or terminology.

  - *FMoW (Functional Map of the World)* (Christie et al., 2018): The FMoW dataset is a large-scale remote sensing collection of approximately 119,000 satellite images gathered between 2002 and 2017 annotated with land-use classes (e.g., airport, forest, hospital). Changes in infrastructure and land use create potential temporal shifts in the data distribution.

Table E.2: Summary of Model Architectures and Hyperparameters

| Dataset | Task | Model | Optimizer | LR | Batch Size | Epochs/Iterations | $\delta$ |
|---|---|---|---|---|---|---|---|
| NewsRoom | Language Modeling | T5 | AdamW | $8 \times 10^{-4}$ | 16 | 1 epoch | 12 months |
| NewsRoom | News Summarization | T5 | AdamW | $8 \times 10^{-4}$ | 16 | 3 epochs | 12 months |
| Yearbook | Gender Prediction | 4-layer CNN | Adam | $1 \times 10^{-3}$ | 32 | 300 iterations | 10 years |
| HuffPost | Headline Topic Classification | DistilBERT | AdamW | $2 \times 10^{-5}$ | 32 | 1000 iterations | 3 years |
| ArXiv | Paper Subject Prediction | DistilBERT | AdamW | $2 \times 10^{-5}$ | 64 | 1000 iterations | 6 years |
| FMoW | Image Classification | Densenet-121 | Adam | $1 \times 10^{-4}$ | 64 | 500 iterations | 6 years |

# D    METRICS FOR TEMPORAL GENERALIZATION

To evaluate how well a model, trained at a specific time $t$, generalizes to future time intervals, we use the $\delta$-forward transfer (FWT) framework (Lopez-Paz and Ranzato, 2017; Yao et al., 2022a). Let $T_{\text{train}}$ be the set of training timestamps for which forward transfer is evaluated. Given the sequence of $T$ total datasets $\{\mathcal{D}_t\}_{t=1}^{T}$ (as defined previously), this set of training timestamps is $T_{\text{train}} = \{t \mid 1 \leq t \leq T - \delta\}$. Let $\mathcal{M}_{t,j}$ denote the performance (e.g., accuracy) of the model associated with training timestamp $t \in T_{\text{train}}$ (e.g., $\theta_t$), when evaluated on data from a future timestamp $j$. The average and worst-case forward transfer metrics are defined as follows.

**Average FWT** ($\text{Avg}_{\text{FWT}}$) aggregates the performance on all valid future datasets within a $\delta$-horizon:

$$\text{Avg}_{\text{FWT}} = \frac{1}{N_{\text{eval}}} \sum_{t \in T_{\text{train}}} \sum_{k=1}^{\delta} \mathcal{M}_{t,t+k}, \tag{14}$$

where $t + k$ denotes the $k$-th future timestamp relative to $t$. $N_{\text{eval}}$ is the total number of $\mathcal{M}_{t,t+k}$ terms included in the sum (i.e., the count of valid $(t, t+k)$ pairs for which performance is measured within the specified horizon). If for every $t \in T_{\text{train}}$, all $\delta$ future evaluation points $(t+1, \ldots, t+\delta)$ are available and valid, then $N_{\text{eval}} = |T_{\text{train}}| \cdot \delta$.

**Worst-case FWT** ($\text{Worst}_{\text{FWT}}$) captures the worst-case performance of each model among its future evaluations within the $\delta$-horizon. This is then averaged over all training timestamps in $T_{\text{train}}$. Assuming $\mathcal{M}$ represents accuracy (where higher is better), we can write it formally as follows:

$$\text{Worst}_{\text{FWT}} = \frac{1}{|T_{\text{train}}|} \sum_{t \in T_{\text{train}}} \min_{k \in \{1, \ldots, \delta\}} \mathcal{M}_{t,t+k}. \tag{15}$$

The $\min$ operation is over the set of $k$ future steps $\{1, \ldots, \delta\}$. If, for a given $t$, $\mathcal{M}_{t,t+k}$ is not available for all $k$ in this range (e.g., if $t + k > T$), the $\min$ should be taken over the subset of these steps for which $\mathcal{M}_{t,t+k}$ is validly defined. If the performance metric $\mathcal{M}$ were an error rate (where lower is better), the $\min$ operator would be replaced by $\max$.

Unlike traditional continual learning, which focuses on mitigating catastrophic forgetting (i.e., maintaining performance on previously learned tasks), our focus here is on forward transfer: assessing how well models trained on past data generalize to future, unseen time horizons.

# E    HYPER-PARAMETERS AND MODEL ARCHITECTURES

This section details the model architectures and hyperparameters used for our experiments. For the NewsRoom dataset, our setup largely follows Nylund et al. (2024), while for the WILDS-Time datasets (Yearbook, HuffPost, ArXiv, and FMoW), we adhere to the experimental configurations outlined in Yao et al. (2022a), unless specified otherwise. A summary is provided in Table E.2, with detailed descriptions following.

## E.1    GENERAL HYPER-PARAMETERS AND MODEL ARCHITECTURES

**NewsRoom dataset**    For the NewsRoom dataset (Grusky et al., 2018), we used T5-small with 70 million parameters and T5-large with 770 million parameters (Raffel et al., 2020). Both models were trained with the AdamW optimizer (Loshchilov and Hutter, 2019) using a learning rate of

$8 \times 10^{-4}$, a batch size of 2, and 8 gradient accumulation steps, resulting in an effective batch size of 16. For language modeling, we used the LM adaptation objective (Lester et al., 2021). The T5-large model was fine-tuned using Low-Rank Adaptation (LoRA (Hu et al., 2022)) with parameters $r = 8$, $\alpha = 32$, and dropout=0.1. LoRA was applied to the query (q) and value (v) attention modules. We used one epoch of training for language modeling and three epochs for downstream tasks.

**Yearbook dataset** For experiments on the Yearbook dataset (Ginosar et al., 2015), we used a 4-layer Convolutional Neural Network (CNN). The model was trained using the Adam optimizer (Kingma, 2014) with a learning rate of $1 \times 10^{-3}$ and a batch size of 32. For each timestamp, the network was trained for 300 iterations and evaluated for gender prediction for ten subsequent years.

**HuffPost dataset** Experiments on the HuffPost dataset (Misra and Grover, 2021) were conducted using an uncased DistilBERT model (Sanh et al., 2019), augmented with a fully-connected classification layer. The AdamW optimizer (Loshchilov and Hutter, 2019) was used for training, with a learning rate of $2 \times 10^{-5}$ and a batch size of 32. Models were trained for 1000 iterations for each timestamp. Performance was evaluated based on predictions for the three years following each timestamp.

**ArXiv dataset** The experimental setup for the ArXiv dataset (Clement et al., 2019) mirrored that of the HuffPost dataset in terms of model architecture, utilizing an uncased DistilBERT model (Sanh et al., 2019) with an appended fully-connected classification layer. Training was performed using the AdamW optimizer (Loshchilov and Hutter, 2019) with a learning rate of $2 \times 10^{-5}$. For this dataset, a batch size of 64 was used, and the model was trained for 1000 iterations. Evaluation was conducted on data from the next six years.

**FMoW dataset** For the FMoW dataset (Christie et al., 2018), we used a Densenet-121 architecture (Huang et al., 2017), which was pre-trained on the ImageNet dataset. The Adam optimizer was used to train the model with a learning rate of $1 \times 10^{-4}$ and a batch size of 64. The training process consisted of 500 iterations. Notably, no L2 regularization was applied during the training of this model. Evaluation was performed on data corresponding to the six subsequent years.

### E.2 METHOD-SPECIFIC HYPER-PARAMETERS

**Method details.** We provide a description of the methods and their hyper-parameters below:

- **Empirical risk minimization (ERM)** in our sequential setting involves training the model at each time $t$ on the cumulative union of all data observed from the start up to time $t$.
- **Invariant risk minimization (IRM)** (Arjovsky et al., 2019) attempts to discover an invariant predictor by learning a data representation where the classifier is consistent across all training domains. The goal is to capture causal mechanisms that are stable across environments while disregarding spurious correlations. We employ the penalty-based approximation with a penalty of $1.0$ for our experiments (Arjovsky et al., 2019; Yao et al., 2022a).
- **Group distributionally robust optimization (GroupDRO)** (Sagawa et al., 2020) optimizes for the worst-case loss encountered in the data from previous time stamps. We follow the standard implementation from previous works (Sagawa et al., 2020; Yao et al., 2022a).
- **Deep correlation alignment (DeepCORAL)** (Sun and Saenko, 2016) adds a loss term that aligns the second-order statistics of the feature distributions between data from a source time period and a target time period. We follow the implementation from prior works (Sun and Saenko, 2016; Yao et al., 2022a) and use a CORAL penalty of $0.9$ for all datasets.
- **Averaged gradient episodic memory (A-GEM)** (Chaudhry et al., 2019) is replay-based continual learning method that uses an episodic memory to store a small number of representative examples from past timestamps. Following prior works (Yao et al., 2022a), a buffer size of 1000 was used for all datasets
- **Elastic weight consolidation (EWC)** (Kirkpatrick et al., 2017) is a regularization based continual learning method that uses a regularization term to penalize significant alterations to model parameters, anchoring the model to previously learned knowledge about past dynamics. A regularization strength of $0.5$ was used across all datasets (Yao et al., 2022a).

For the downscaling method, we used $\alpha = 0.956892$ for T5-Small (for all timestamps in both language modeling and news summarization tasks on the NewsRoom dataset) and $\alpha = 0.988181$

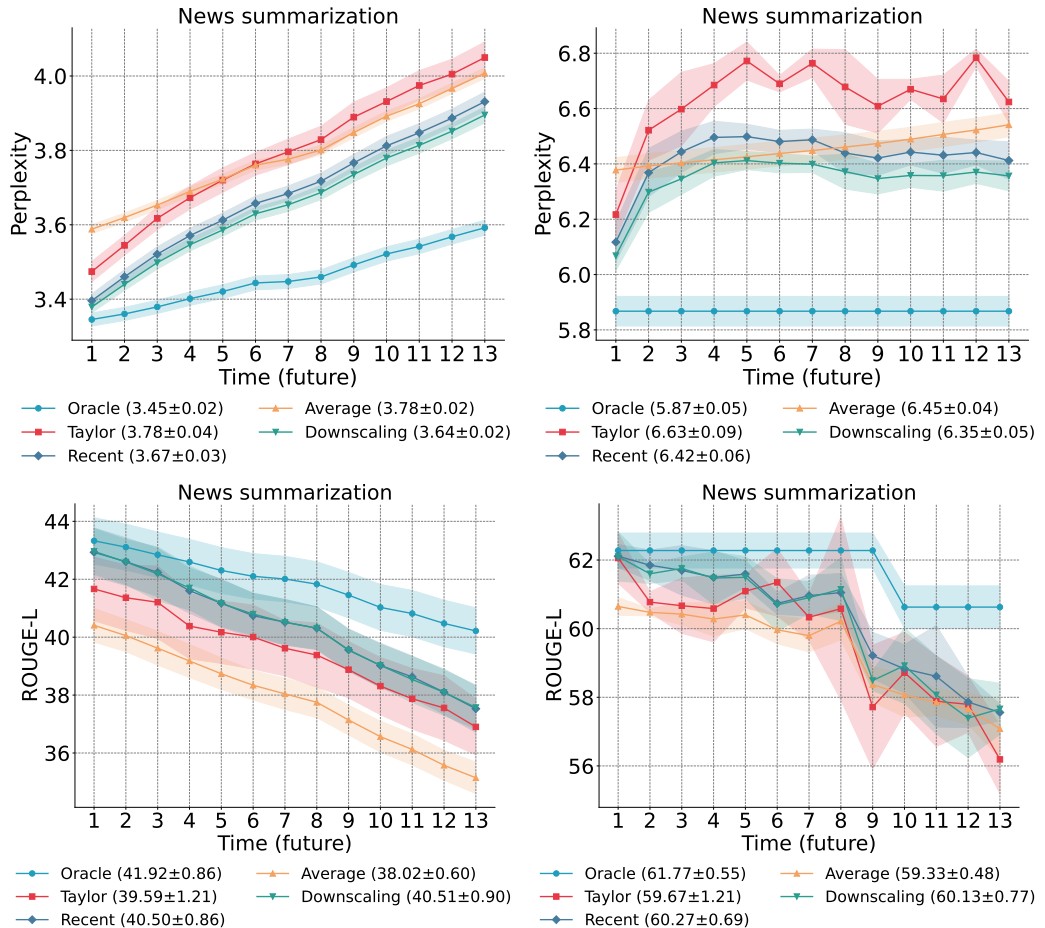

Figure F.11: **Perplexity and ROUGE-L comparison for T5-large model.** We contrast average-case performance (left panels) with worst-case performance (right panels) over future time steps (x-axis). Effective summarization is marked by lower Perplexity scores and higher ROUGE-L scores (y-axis). Downscaling is the only method that does not consistently lead to performance decay across both evaluation metrics compared to the recent model.

for T5-Large. These $\alpha$ values were determined via a random grid search over the interval $[0.9, 1.0]$. To obtain the Taylor approximation $\alpha$ in Equation (6), we conducted a random grid search with 30 values sampled from the interval $[-1, 1]$. For Wilds-Time, we found $\alpha = 1.0$ to be consistently optimal for the simulated future for the datasets. Each experiment was run five times to report the mean and standard deviation.

## F  ADDITIONAL EXPERIMENTAL RESULTS

**Results with T5-Large and T5-small model.**   The results for T5-large in Figure F.11 and T5-small in Figure F.12 largely affirm the key trends discussed in the main paper. None of the benchmarked methods including parameter averaging, first-order Taylor expansion and downscaling consistently outperform the recent model. Regarding the evaluation metrics, we contend that ROUGE-L presents considerable limitations for assessing temporal generalization. Its inherent sensitivity to lexical variations, rather than semantic shifts, can be misleading in this context. Many studies show an inconsistent correlation of ROUGE-L with human evaluations across different applications (Goyal et al., 2022; Cohan and Goharian, 2016; Grusky, 2023). It may not adequately capture the nuanced improvements associated with language model scaling, where semantic fidelity often outweighs exact lexical replication (Sellam et al., 2020; Zhang et al., 2020).

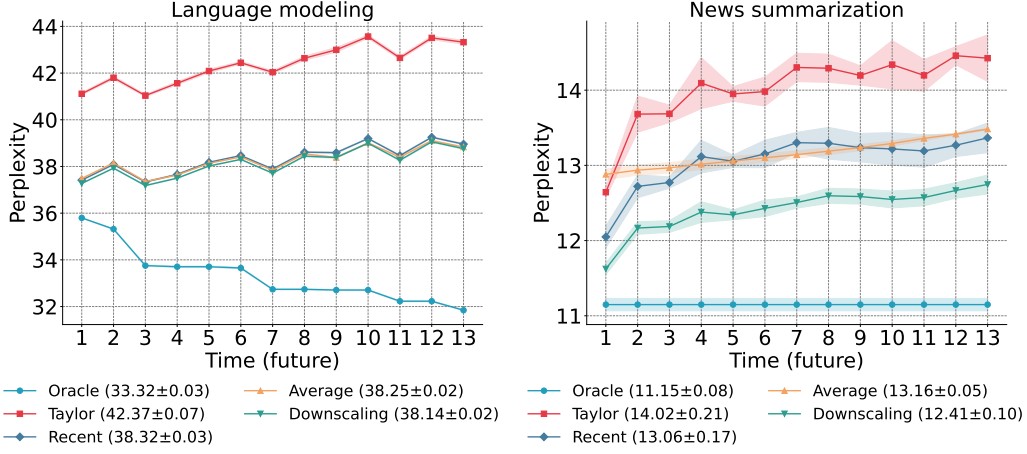

(a) **Worst-case Perplexity for T5-small model.** The left panel shows results for language modeling and the right panel for news summarization. Each method is evaluated over future time steps (x-axis), with lower Perplexity values (y-axis) indicating better performance. Consistent with the average-case performance, we see that only downscaling does not lead to a drop in performance compared to a recent model.

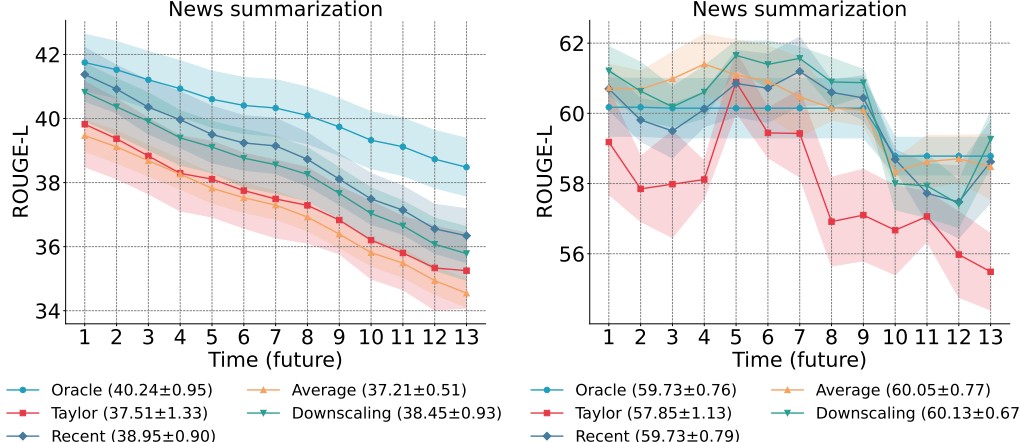

(b) **Average and Worst-case ROUGE-L for T5-small model.** The left panel shows results for language modeling and the right panel for news summarization. Each method is evaluated over future time steps (x-axis), with higher ROUGE-L values (y-axis) indicating better performance.

Figure F.12: Performance evaluation with perplexity and ROUGE-L of the T5-small model using various methods on language modeling and news summarization tasks.

**Yearly results with T5-small model.** We further evaluated interpolation and extrapolation strategies over two future years on the news summarization task using a T5-small model. Consistent with prior monthly results, the Taylor expansion yielded weaker performance, with a significantly higher average perplexity of $9.274 \pm 0.060$ compared to using the most recent model (average perplexity $7.582 \pm 0.019$). Downscaling the parameters of the recent model achieved a comparable average perplexity of $7.562 \pm 0.006$. For the second future year, this scaling approach resulted in a lower perplexity of $7.589 \pm 0.006$ for downscaling compared to $7.661 \pm 0.022$ for the recent model. These findings from multi-year evaluation highlight the limitations of simpler Taylor series-based methods in this context and suggest that appropriate scaling of recent model parameters can be a simple strategy for temporal generalization.

**Additional dimensionality reduction visualizations.** We show the TSNE visualization during language modeling and news summarization with continual learning in Figure F.17 and Figure F.18. We further show UMAP, PCA visualization during language modeling in Figure F.16 and Figure F.19 and news summarization with continual learning in Figure F.20.

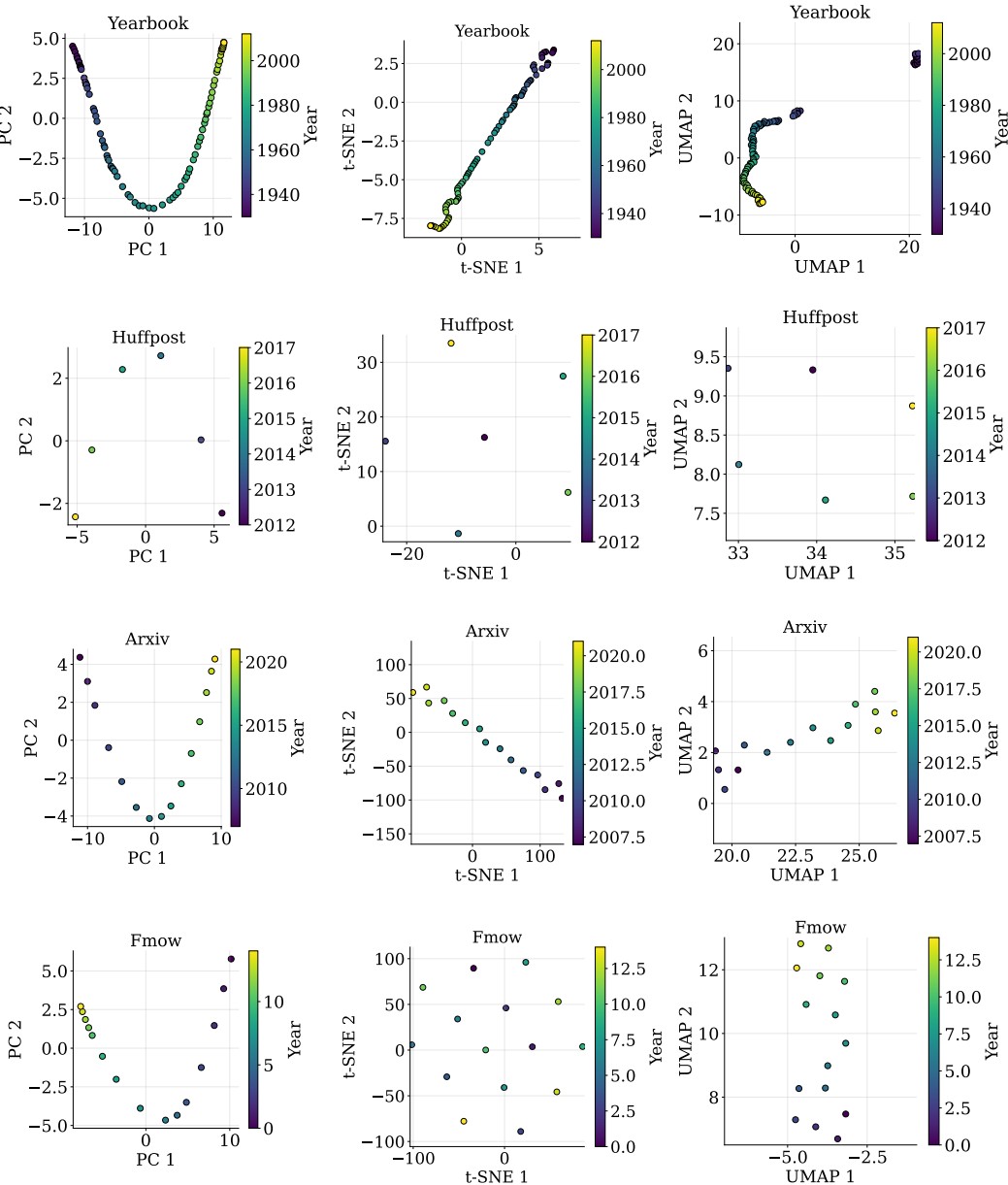

Figure F.13: **Dimensionality reduction of model parameters over time for WILDS-Time datasets.** Each scatter plot shows a 2D projection of model checkpoints using PCA, TSNE and UMAP dimensionality reduction techniques.

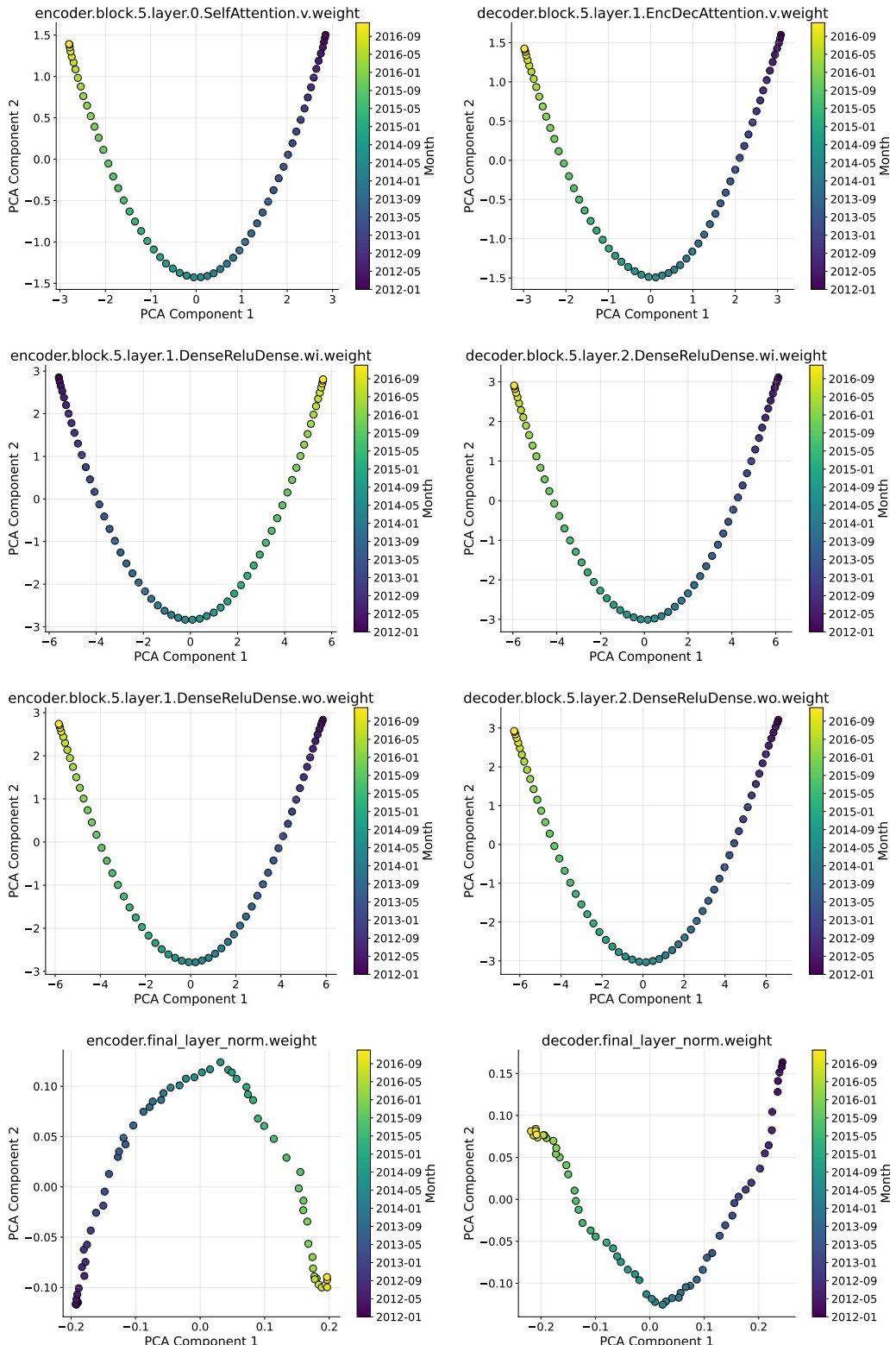

Figure F.14: **Smooth parameter trajectories under continual learning.** PCA of T5-small weights during news summarization with continual learning shows smooth, coherent evolution over time. This contrasts with independently fine-tuned checkpoints (Figure F.15), highlighting how continual learning preserves temporal consistency in parameter space and supports forward transfer.

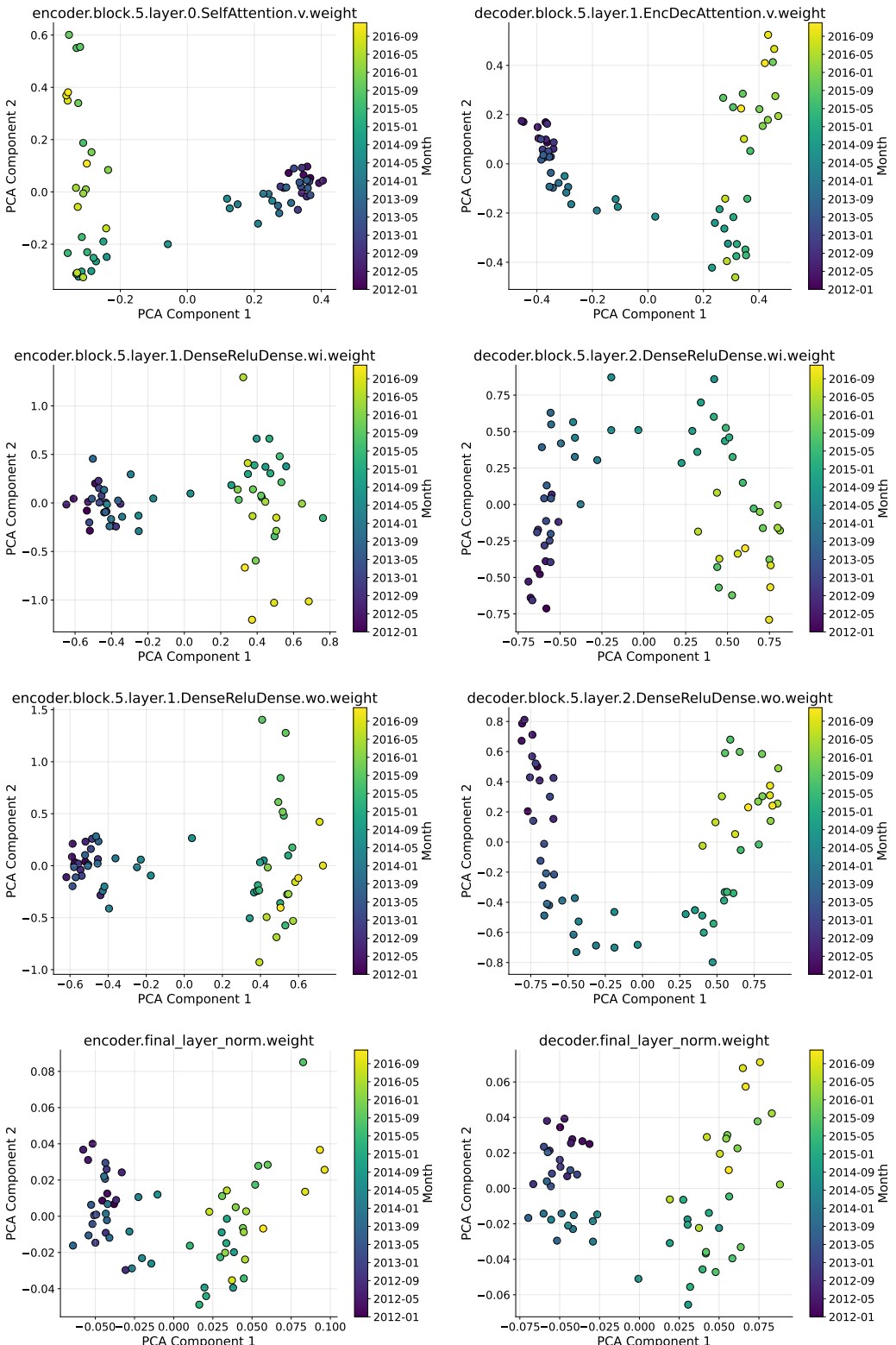

Figure F.15: **Parameter trajectories from independent finetuning.** PCA of T5-small weights during news summarization, where each timestamp model is trained independently from a pre-trained checkpoint. The lack of smoothness across time steps indicates inconsistent parameter evolution and highlights the challenges of extrapolation in the absence of continual learning.

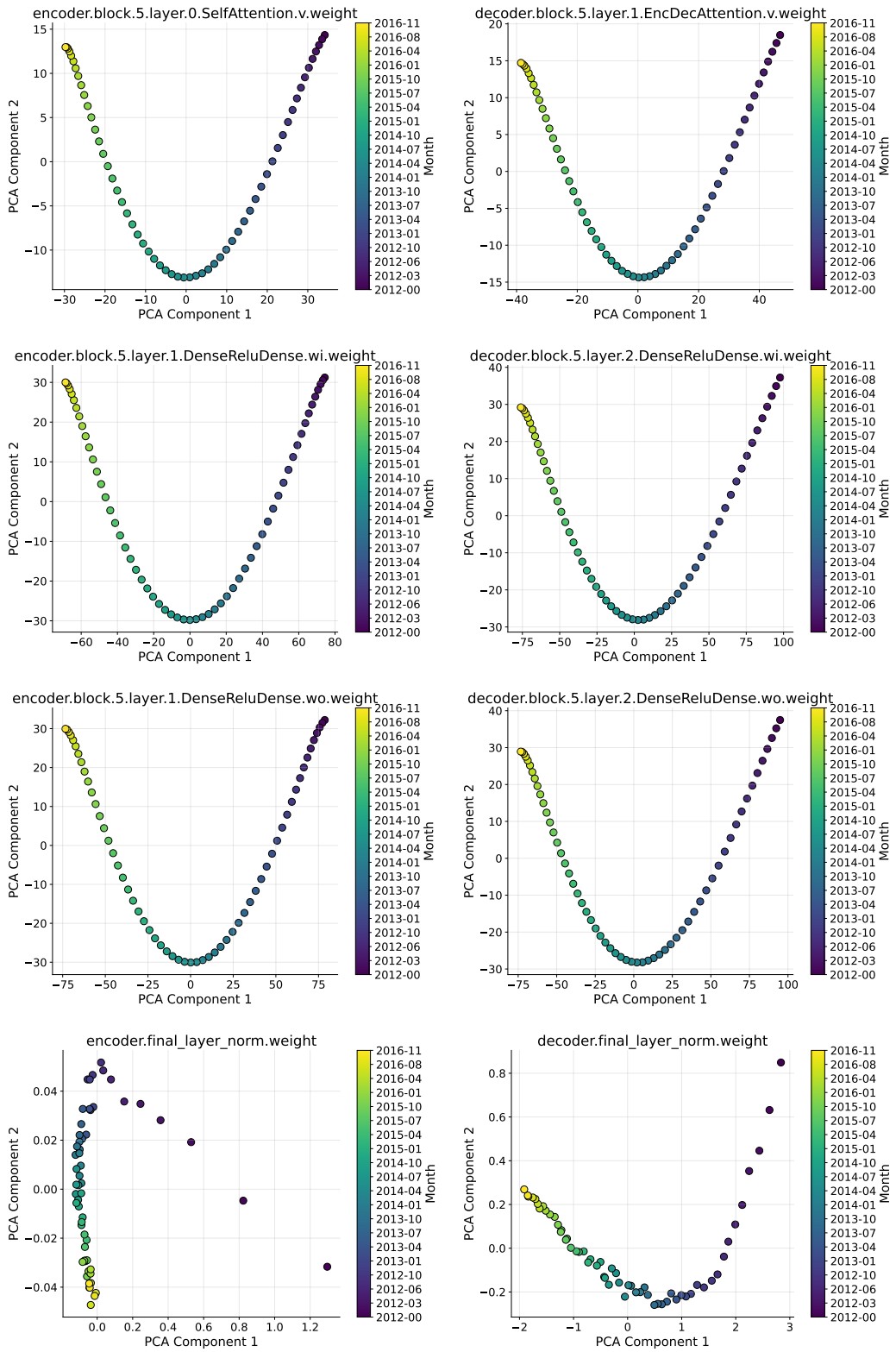

Figure F.16: **Smooth parameter trajectories under continual learning.** PCA of T5-small weights during language modeling with continual learning shows smooth, coherent evolution over time.

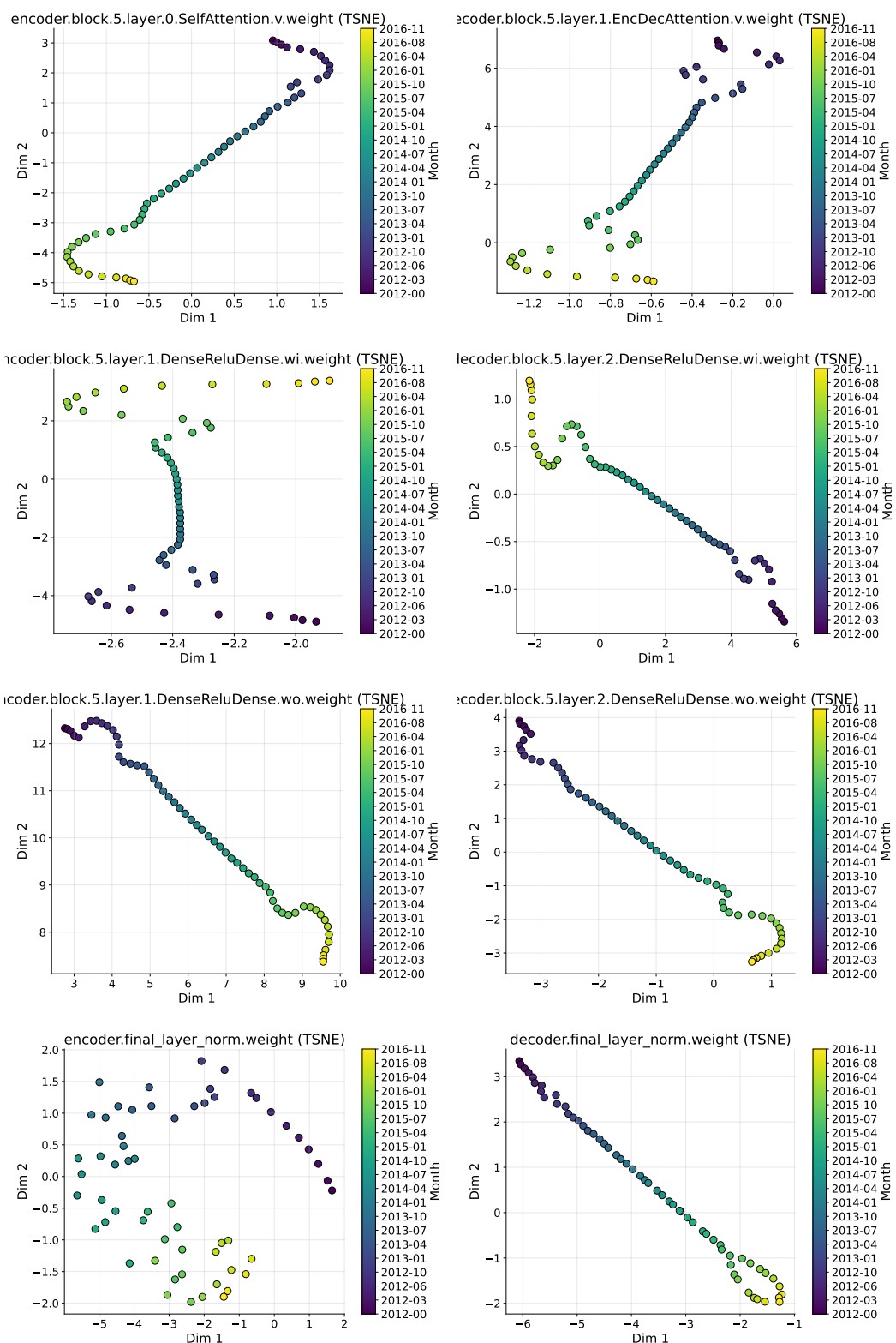

Figure F.17: TSNE of T5-small weights during language modeling with continual learning.

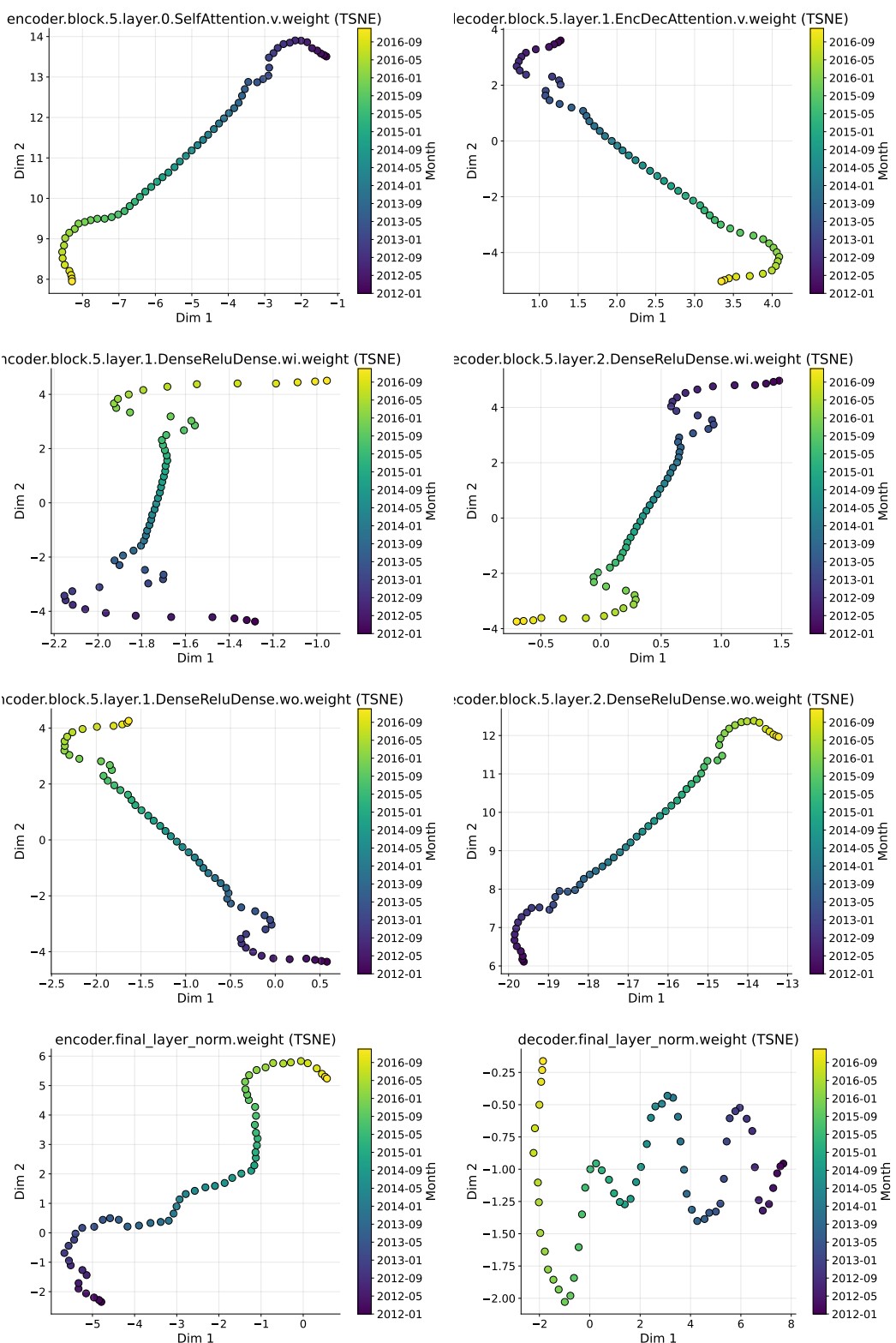

Figure F.18: TSNE of T5-small weights during news summarization with continual learning.

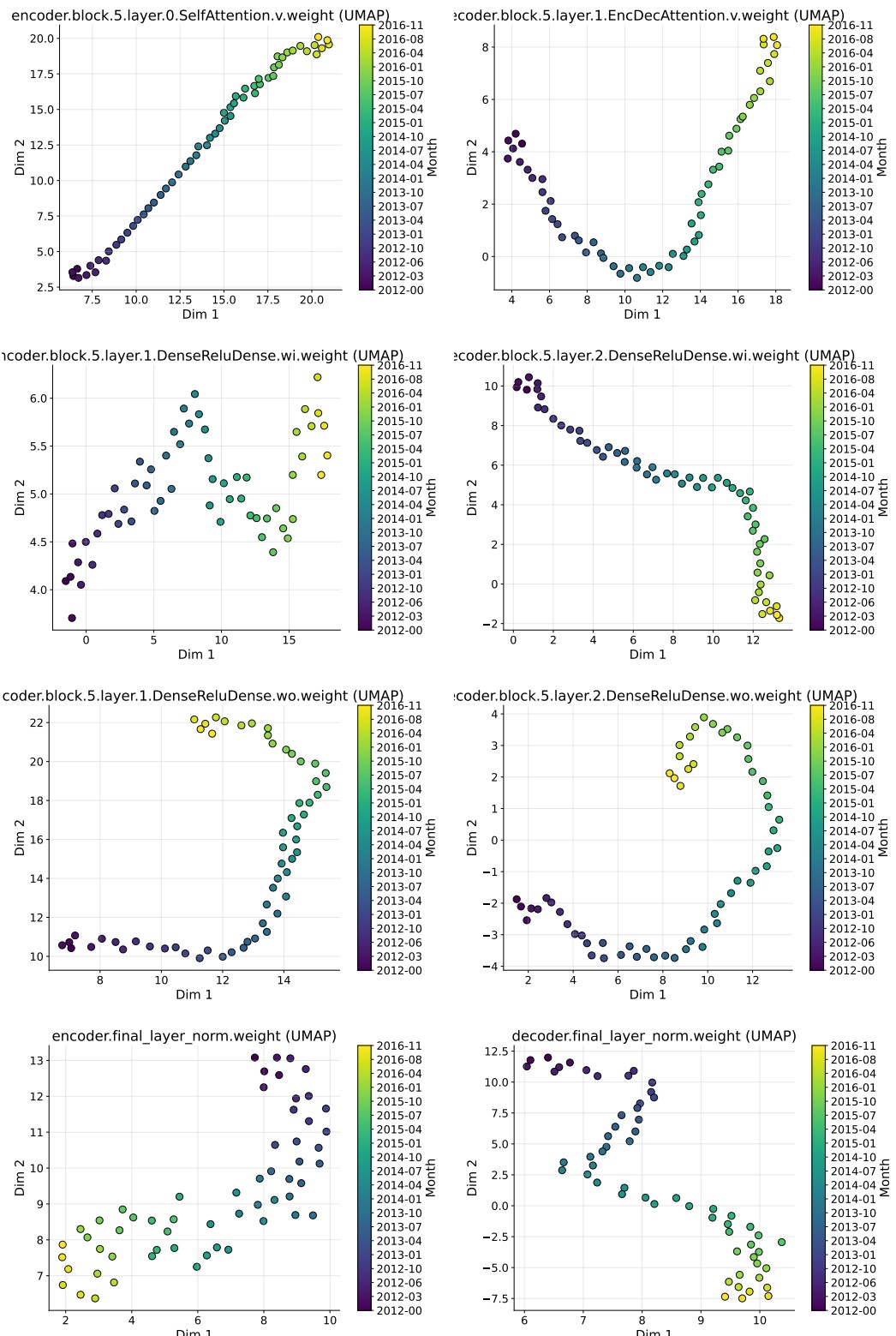

Figure F.19: UMAP of T5-small weights during language modeling with continual learning.

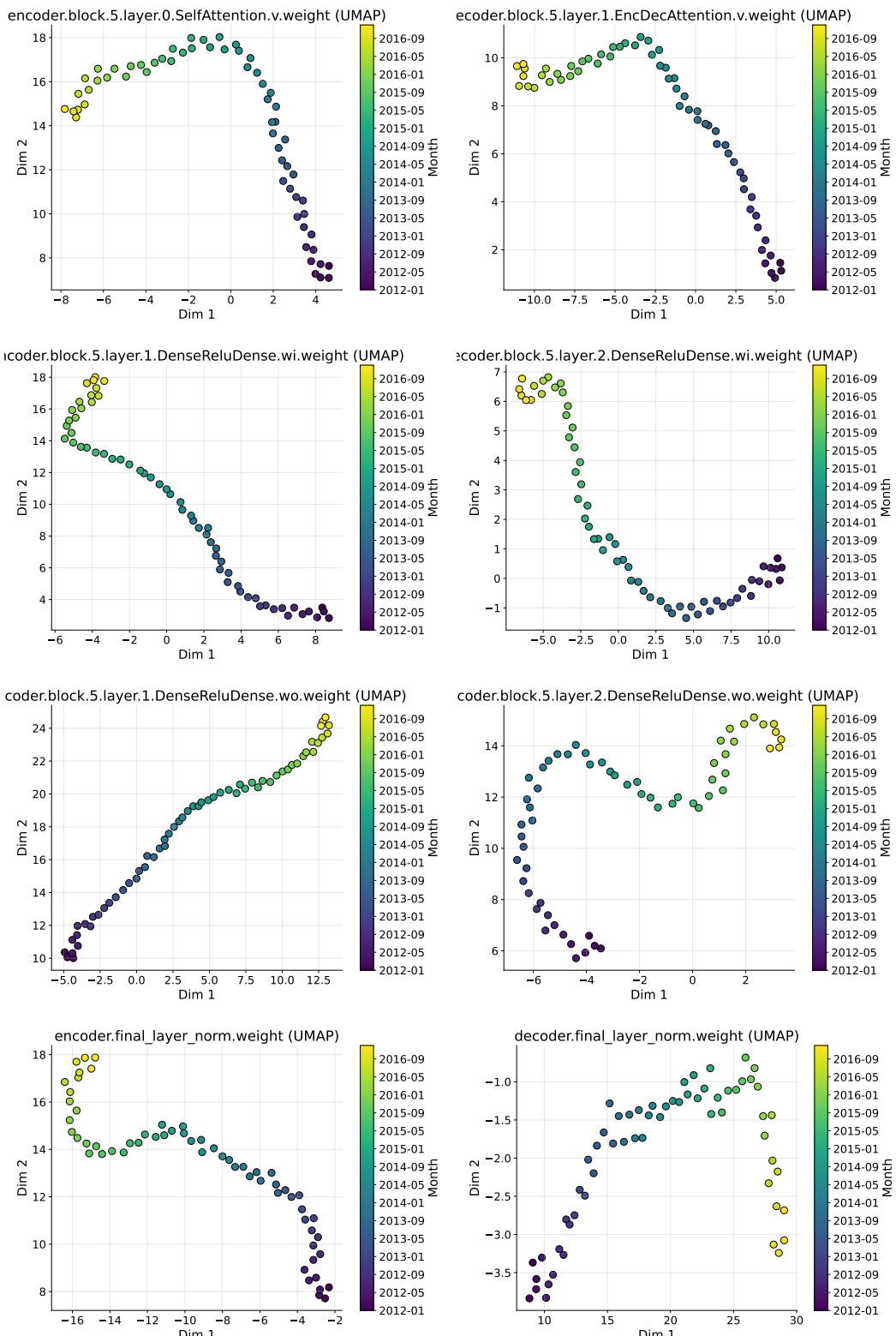

Figure F.20: UMAP of T5-small weights during news summarization with continual learning.

