# OpenReview forum: "Temporal Generalization: A Reality Check"
_ICLR.cc/2026/Conference — ICLR 2026 Poster_

### Official Review · Reviewer_azAb · 2025-10-17

**Soundness:** 4
**Presentation:** 3
**Contribution:** 3
**Rating:** 8
**Confidence:** 3

**Summary:**

This paper deals with the issue of predictions on future data based on past checkpoints, presenting comprehensive benchmarking on various tasks (language modeling on evolving news corpora, news summarization and tag prediction, categorization
of academic papers, land use classification from satellite imagery reflecting changes over several
years, and gender prediction from historical yearbook photos ). Existing benchmarking studies are hampered by unrealistic evaluation settings as they e.g. assume access to future data.
The core hypothesis is that approaches can be broken down into two phenomena: interpolation vs. extrapolation. They adopt a sequential fine-tuning approach.

Their core findings are:
1) No method consistently improves performance over the recent model.
2) Taylor-Series extrapolation underperforms compared to other methods.
3) Continual learning is important.

**Strengths:**

The experiments presented in the paper are extensive and are clearly guided by research questions, which are answered and discussed well.

**Weaknesses:**

There are minor parts of the presentation that could be improved, e.g. spell-checking, using the same font for all figures and the same style for all figure captions. Table 1 is also positioned strangely.

**Questions:**

N/A

---

> ### Author Response · Authors · 2025-11-20
> **Response to Reviewer azAb**
>
> Dear Reviewer azAb,
>
> We appreciate that you found our work to contain extensive experiments with clearly guided and answered research questions. In the updated revision, we conducted a spell-check and fixed the remaining issues highlighted in your review. Table 1 was positioned within the text due to space constraints and we will update it in the final revision. We are further happy to address any remaining issues that you highlight. Thank you again for your time and efforts in reviewing our paper, and we hope that you will consider adjusting your score if you find our response satisfactory.
>
> Thank you,
> Authors

---

### Official Review · Reviewer_z2gD · 2025-10-30

**Soundness:** 4
**Presentation:** 4
**Contribution:** 4
**Rating:** 8
**Confidence:** 3

**Summary:**

This study asks whether ML models can truly generalize to future, unseen data when you only have access to past model checkpoints—no peeking at future data. The authors benchmark two families of methods that operate purely on historical parameters: (i) parameter interpolation (averaging/EMA and a simple downscaling of the latest model), and (ii) parameter extrapolation (first-order, Taylor-style forecasting of weights). Across language modeling and summarization on monthly news, plus yearly tasks like Yearbook, HuffPost, arXiv, and FMoW, they find no approach reliably beats simply deploying the most recent model; notably, downscaling the latest weights is the only method that doesn’t consistently degrade performance.

**Strengths:**

The paper introduces a highly realistic experimental framework that mirrors real-world deployment, strictly prohibiting the use of future data.

It delivers a comprehensive empirical evaluation, testing multiple architectures and datasets with different temporal frequencies and scales.

The authors offer transparent and reproducible experimentation, including open-source code and detailed implementation settings.

The study provides valuable negative evidence, demonstrating that intuitive weight-based strategies can fail dramatically, which grounds future work in reality rather than optimism.

Beyond reporting results, it also diagnoses the causes—linking observed failures to theoretical properties such as non-identifiability and parameter-space discontinuity.

**Weaknesses:**

The investigation remains restricted to parameter-space manipulation, leaving out more adaptive or data-driven temporal methods that could exploit unlabeled future samples.

Some evaluated datasets exhibit limited temporal granularity, reducing the precision with which the authors can analyze temporal drift.

The reliance on historical validation only makes hyperparameter selection brittle and may underestimate the potential of some methods.

While empirically thorough, the paper’s methodological contribution is modest, focusing on benchmarking rather than innovation.

**Questions:**

N/A

---

> ### Author Response · Authors · 2025-11-20
> **Response to Reviewer z2gD**
>
> Dear Reviewer z2gD,
>
> Thank you for your insightful and detailed feedback. We appreciate that you found our work to contain highly realistic and comprehensive experimental evaluation. We address your specific concerns below.
>
> > [W1] The investigation remains restricted to parameter-space manipulation, leaving out more adaptive or data-driven temporal methods that could exploit unlabeled future samples.
>
> There are empirical unsupervised domain adaptation algorithms that can be applied for temporal generalization; however, this is outside the scope of our paper. We focus on temporal generalization with no available knowledge about the future. It would be interesting to conduct this exhaustive study as a followup in the future.
>
> ---
>
> > [W2] Some evaluated datasets exhibit limited temporal granularity, reducing the precision with which the authors can analyze temporal drift.
>
> The lack of fine-grained public datasets for temporal generalization remains a major challenge. The NewsRoom dataset is among the few that has a finer temporal resolution. We would appreciate it if the reviewer could suggest specific datasets that meet these criteria for future work.
>
> ---
>
> > [W3] The reliance on historical validation only makes hyperparameter selection brittle and may underestimate the potential of some methods.
>
> The brittleness of hyperparameter selection is precisely the core challenge of temporal generalization and one of the reasons prior work appeared to observe performance improvements. We hope that our paper will motivate future work to make assumptions about changing data rather than using future data to choose hyperparameters.
>
> ---
>
> > [W4]  While empirically thorough, the paper’s methodological contribution is modest, focusing on benchmarking rather than innovation.
>
> While developing a novel method that solves the problem is the end goal, we feel that this cannot be accomplished without a thorough understanding of the underlying problem and the deficiencies in current approaches. Unfortunately, in the domain of temporal generalization, there is a significant gap in this direction due to the lack of rigorous benchmarks. Consequently, this paper evaluates a wide range of standard approaches, shows that they perform poorly in practice, and seeks to shed light on why. We thus hope that our large-scale benchmarking will inspire future innovations evaluated under realistic deployment scenarios for temporal generalization.
>
> ---
>
> Thank you again for your time and efforts in reviewing our paper, and we hope that you will consider raising your score if you find our response satisfactory.
>
> Thank you,
> Authors

---

> ### Comment · Reviewer_z2gD · 2025-11-20
>
> thanks for your reply, for [W2], do MIMIC-iii and MIMIC-IV have a finer temporal resolution?

---

> ### Author Response · Authors · 2025-11-21
> **Thanks for your response**
>
> Dear Reviewer z2gD,
>
> For the de-identification of MIMIC-III and MIMIC-IV, all dates were shifted using a patient-level offset, where each patient was assigned a random date shift. This makes any temporal split of the dataset extremely challenging.
>
> Thank you,
> Authors

---

### Official Review · Reviewer_KbVf · 2025-11-01

**Soundness:** 3
**Presentation:** 3
**Contribution:** 3
**Rating:** 6
**Confidence:** 3

**Summary:**

The paper studies temporal generalization — the ability of models trained on past data (and only past data) to perform well on future, potentially shifted data. The authors point out that many recent papers implicitly assume that, given a sequence of checkpoints, one can interpolate or extrapolate those weights to obtain parameters that work better on future data, even when the future distribution is unknown. The central claim of the paper is negative but important: under a realistic, “no access to the future” setting with streaming / continual updates, none of the evaluated interpolation (uniform / EMA merging, recent model, norm-downscaling) or extrapolation (first-order Taylor on parameter trajectory) methods consistently beats the trivial baseline of “just use the latest model. The only method that consistently avoids hurting performance is a simple downscaling of the latest model (interpolating toward the origin), which the authors motivate via parameter-norm growth and sharper minima over time.

**Strengths:**

1. The paper is very explicit about the setting: streaming / continual fine-tuning, no access to future unlabeled or labeled data, and hyperparameters must be chosen from the past. This closes an often-overlooked loophole in earlier temporal/DG works that quietly validated on future data.
2. The experiments span: (i) monthly news language modeling and summarization with T5 (showing clear temporal decay and the salience of downscaling), and (ii) WILDS-Time datasets (Yearbook, HuffPost, arXiv, FMoW), where no single method dominates. This makes the main negative result much more convincing than if it were shown on one dataset.
3. The paper separates parameter interpolation (merging, recent, downscaling) from parameter extrapolation (Taylor step using the most recent difference).
4. The observation that the optimal extrapolation factor 𝛼 often drifts below 1 or even becomes negative (meaning “don’t extrapolate; sometimes go backward”) is a strong piece of evidence that parameter trajectories of large models are not sufficiently smooth or aligned over time to support simple Taylor-based prediction.
5. The finding that downscaling the latest model is the only option that does not reliably hurt, and that this correlates with steadily growing parameter norms under continual learning, is a small but real contribution that practitioners can immediately try.

**Weaknesses:**

1. The paper explains why interpolation/extrapolation is hard via non-identifiability, non-convexity, and noisy parameter trajectories, but these are largely qualitative or synthetic-illustration arguments; there is no general impossibility statement of the form “under assumptions X, any future-agnostic checkpoint-only method cannot beat the latest model by more than 𝜖”. A lightweight formalization, even under a stylized model, would sharpen the negative result.
2. The paper itself shows that if you don’t do sequential fine-tuning (i.e. if you reinit from a pretrain every time), all methods degrade much more (Fig. 6). That means the key negative result is partly conditional on the positive result that CL keeps consecutive checkpoints close. This is important, but it also means the conclusions are strongest for CL-like pipelines, weaker for other update regimes.
3. The paper does the right thing — tune 𝛼 only on past data — but for some methods (esp. extrapolation) the chosen 𝛼 fluctuates sharply over time (Fig. 5). In practice, this means users would have to re-tune at every step, which undercuts the appeal of “cheap temporal generalization.” A small ablation on “fixed  𝛼 over long horizons” vs “retuned 𝛼 ” would have made this clearer.

**Questions:**

1. You show that parameter norms steadily increase under CL and that downscaling helps. Did you try layerwise or modulewise downscaling (e.g. scale only attention / only FFN / only classifier) and, if so, was the benefit still present? Intuitively, temporal brittleness might concentrate in the last layers that overfit to very recent tokens / labels.
2. Can you tell whether the sign flip correlates with which dataset was seen last (e.g. news vs scientific vs vision)? In other words, is the problem that parameter trajectories are globally noisy, or that mixing modalities / tasks breaks linearity?
3. Fig. 6 clearly shows CL helps. But you fine-tune from θt−1 on all datasets. What happens if the per-step compute is reduced (e.g. 10% of the usual steps, or LoRA-style adapters only)? Do interpolation/extrapolation fail immediately because checkpoints become less aligned, or do they degrade more gracefully?

---

> ### Author Response · Authors · 2025-11-20
> **Response to Reviewer KbVf**
>
> Dear Reviewer KbVf,
>
> Thank you for your insightful and detailed feedback. We appreciate that you found our work to close a loop hole in existing works with convincing results and a practical contribution. We address your specific concerns below.
>
> > [W1] There is no general impossibility statement of the form “under assumptions X, any future-agnostic checkpoint-only method cannot beat the latest model by more than 𝜖”. A lightweight formalization, even under a stylized model, would sharpen the negative result.
>
> Having established that most existing algorithms do not work for temporal generalization when future data is unavailable, we agree that formalizing these limitations is a natural next step. However, a general impossibility result is not as straightforward as the reviewer suggests. To obtain meaningful guarantees, we need to find the tightest set of assumptions under which this statement can be made, which we believe is a great direction for a follow up study.
>
> ---
>
> > [W2] The paper itself shows that if you don’t do sequential fine-tuning (i.e. if you reinit from a pretrain every time), all methods degrade much more (Fig. 6). That means the key negative result is partly conditional on the positive result that CL keeps consecutive checkpoints close. This is important, but it also means the conclusions are strongest for CL-like pipelines, weaker for other update regimes.
>
> Our conclusion holds for all update regimes, as all methods degrade on future data regardless of how the checkpoints are obtained. Interpolation and extrapolation from a pre-trained model are even more challenging and in some cases infeasible due to the challenges highlighted in Section 4 of the paper.
>
> ---
>
> >  [W3] The paper does the right thing — tune $\alpha$ only on past data — but for some methods (esp. extrapolation) the chosen $\alpha$ fluctuates sharply over time (Fig. 5). In practice, this means users would have to re-tune at every step, which undercuts the appeal of “cheap temporal generalization.” A small ablation on “fixed $\alpha$ over long horizons” vs “retuned $\alpha$ ” would have made this clearer.
>
> Because of this exact limitation of extrapolation, we recommend downscaling. Due to $\alpha$ being negative, downscaling is a more natural approach for all settings. Downscaling is a cheap method in terms of compute because we use the same alpha for all time-stamps. We did not find $\alpha$ to change much across time-stamps. We provide these details in Section E.2, downscaling for T5-small and T5-large uses $\alpha$ to be $0.956892, 0.988181$ and 1.0 for Wilds-time. These values were obtained with a random grid search between range $[0.9, 1.0]$.
>
> ---
>
> > [Q1 You show that parameter norms steadily increase under CL and that downscaling helps. Did you try layerwise or modulewise downscaling (e.g. scale only attention / only FFN / only classifier) and, if so, was the benefit still present? Intuitively, temporal brittleness might concentrate in the last layers that overfit to very recent tokens / labels.
>
> The decision of choosing the downscaling parameters for different layers and modules is non-trivial. Particularly, the choice of layers and downscaling parameters based on the assumptions of changes across time is extremely difficult and an important direction for future. In Table 1, we made an attempt to learn the downscaling coefficients and extrapolation parameters for the entire model but found no performance benefit.
>
> ---
>
> > [Q2] Can you tell whether the sign flip correlates with which dataset was seen last (e.g. news vs scientific vs vision)? In other words, is the problem that parameter trajectories are globally noisy, or that mixing modalities / tasks breaks linearity?
>
> While we did not evaluate mixing multiple modalities in this work, the relationship between subsets within a modality remains unclear. It is possible that, with an appropriately chosen subset of the dataset, some structure allows the model to generalize temporally. However, identifying such a subset in advance is difficult and remains an open problem.
>
> ---
>
> > [Q3] Fig. 6 clearly shows CL helps. But you fine-tune from $\theta_{t-1}$ on all datasets. What happens if the per-step compute is reduced (e.g. 10% of the usual steps, or LoRA-style adapters only)? Do interpolation/extrapolation fail immediately because checkpoints become less aligned, or do they degrade more gracefully?
>
> It is possible that certain training setups or parameter subsets could exhibit better temporal generalization under reduced per-step compute or restricted updates. However, identifying which subset of parameters should be updated, without access to future data, remains challenging and is an interesting direction for future work.
>
> ---
>
> Thank you again for your time and efforts in reviewing our paper, and we hope that you will consider raising your score if you find our response satisfactory.
>
> Thank you,
> Authors

---

### Official Review · Reviewer_Zdv8 · 2025-11-09

**Soundness:** 3
**Presentation:** 3
**Contribution:** 3
**Rating:** 6
**Confidence:** 3

**Summary:**

Due to inaccurate prediction on unseen future data by Machine Learning (ML) models, the authors investigate under what conditions models can achieve such a generalization when depending on past data. The authors explore two Primary approaches: Parameter Interpolation and Parameter Extrapolation. Furthermore, several methods were benchmarked on a set of temporal tasks, including language modeling, news summarization, tag prediction, academic paper categorization, and land use classification.

**Strengths:**

1. It presents a large-scale, systematic evaluation across multiple tasks (language modeling, summarization, paper categorization, land use classification, and photo gender prediction) and datasets (NewsRoom, Wilds-Time, arXiv, HuffPost, FMoW, Yearbook).

2. The paper makes a contribution by showing that none of the proposed or existing methods reliably outperform a simple "most recent model" baseline.
3. Through parameter norm analysis and dimensionality reduction (PCA/UMAP), the paper provides intuitive interpretations of why downscaling helps.
4. The authors emphasize reproducibility and maintaining transparent reporting of hyperparameters and evaluation setups.

**Weaknesses:**

1. The paper primarily evaluates existing approaches (parameter interpolation, extrapolation, downscaling) rather than introducing a new algorithm or theoretical framework.

**Questions:**

1. What was the data size for each time stamp when using different datasets for different methods?
2. While the paper includes an Oracle model in Figures 2, its definition and implementation are not clearly explained in the main text.

---

> ### Author Response · Authors · 2025-11-20
> **Response to Reviewer Zdv8**
>
> Dear Reviewer ZDv8,
>
> Thank you for your insightful and detailed feedback. We appreciate that you found our work to have a large-scale systematic evaluation with intuitive interpretation. We address your specific concerns below.
>
> > [W1] The paper primarily evaluates existing approaches (parameter interpolation, extrapolation, downscaling) rather than introducing a new algorithm or theoretical framework.
>
> Though the paper does not introduce a new algorithm, the ramifications of the presented study are equally compelling. Understanding the nature of the problem and the deficiencies in the current approaches is a vital stepping stone to developing any new algorithm that works. Unfortunately, in the domain of temporal generalization, there is a significant gap in this direction due to the lack of rigorous benchmarks. Consequently, this paper evaluates a wide range of standard approaches, shows that they perform poorly in practice, and sheds light on why they perform poorly. The objective of this research is to highlight that no single algorithm works across all datasets, as each dataset requires different assumptions about how its distribution changes over time. Identifying these assumptions for each dataset is non-trivial due to the inherent unpredictability of future data. A new algorithm, thereby, remains an open problem for future work.
>
> ---
>
> > [Q1] What was the data size for each time stamp when using different datasets for different methods?
>
> NewsRoom contains 1043, 118 and 109 examples for training, validation and testing for summarization. For Wilds-Time, please refer to Table 3, 4, 6, 7 for Yearbook, FMoW, HuffPost and ArXiv datasets respectively in the Wilds-Time paper [1]. The same size of datasets was used for all methods.
>
> ---
>
> > [Q2] While the paper includes an Oracle model in Figures 2, its definition and implementation are not clearly explained in the main text.
>
> Oracle contains complete knowledge of the future, trained and evaluated on data from time $t$. The details have also been updated in Figure 2 of the revision.
>
> ---
>
> Thank you again for your time and efforts in reviewing our paper, and we hope that you will consider raising your score if you find our response satisfactory.
>
> Thank you,
> Authors
>
> [1] Yao et al. Wild-Time: A Benchmark of in-the-Wild Distribution Shift over Time

---

### Author Response · Authors · 2025-12-03
**Rebuttal summary**

Dear Area Chair and reviewers,

We are thankful to the reviewers for their feedback. All reviewers found our work to be highly realistic, closing a loophole in existing works with a large-scale systematic evaluation and intuitive interpretation.

We want to reiterate that temporal generalization lacks understanding due to the absence of rigorous benchmarks and evaluations. Our paper addresses this gap by evaluating a wide range of standard approaches, showing that they perform poorly in practice and providing insights into why this occurs. We highlight that no single algorithm works across all datasets, as each dataset requires different assumptions about how its distribution changes over time.

The future is unpredictable and we hope that our work will be a vital stepping stone for developing future algorithms. We encourage future exploration into reliable hyper-parameter selection, the curation of fine-temporal benchmarks and the identification of the dataset-specific assumptions that are essential for temporal generalization.

Thank you,
Authors

---

### Meta-Review · Area_Chair_sNFi · 2025-12-03

**Summary:**

The authors study temporal problems where the data distribution changes over time. Their finding shows that, without access to future data, the most recently trained model works better than using combinations of past parameter sets (i.e., intra- and extrapolated, downscaling). While the results appear anti-climatic, they are generated using a diverse set of problems, data types and data sets and could be considered lessons learned in general sequential/temporal prediction problems.

**Reviewer Concerns:**

Reviewers criticise the lack of a technical contribution but the paper does not strive for technical novelty. Instead, the paper aims to empirically study how to approach sequential prediction problems under distribution shift in practice. I am under the impression that the authors clarified most of the questions and concerns raised by the reviewers in their rebuttal.

**Reviewer Scores:**

Since all reviewers already agree to accept the paper, the rebuttal could possibly have a slight positive effect on the actual ratings but would not change the general picture. Since the paper is rather an application paper, I would follow the reviewers and accept the paper as poster.

---

### Decision · Program_Chairs · 2026-01-26

Accept (Poster)